# CONFIDENCE CALIBRATION IN VISION-LANGUAGE-ACTION MODELS

## ABSTRACT

Trustworthy robot behavior requires not only high levels of task success but also that the robot can reliably quantify how likely it is to succeed. To this end, we present a first-of-its-kind study of confidence calibration in vision-language-action (VLA) foundation models, which map visual observations and natural language instructions to low-level robot motor commands. We examine how task success relates to calibration error and how calibration evolves over time, and introduce two lightweight techniques to remedy the miscalibration we observe: prompt ensembles and action-wise Platt scaling. Our aim in this study is to begin to develop the tools and conceptual understanding necessary to render VLAs both highly performant and highly trustworthy via reliable uncertainty quantification.

## 1 INTRODUCTION

*Confidence calibration*, or the degree to which a model's predicted probabilities reflect the true likelihood of its predictions being correct, is a cornerstone of reliable machine learning systems (Guo et al., 2017). When a model is well-calibrated, downstream decision-makers (e.g., humans, planning algorithms, or safety monitors) can trust that a confidence estimate of 95% implies that the predicted outcome will occur roughly 95% of the time (see Figure 1). Mismatches between confidence and actual outcomes can have severe negative consequences in high-stakes settings such as medical diagnosis or autonomous driving, motivating a rich literature on improving calibration in deep learning models (Minderer et al., 2021; Lakshminarayanan et al., 2017; Gal & Ghahramani, 2016; Tian et al., 2023; Lin et al., 2024; Kadavath et al., 2022).

Recently, the field of robotics has embraced a new class of *vision-language-action* (VLA) foundation models (Zitkovich et al., 2023; Kim et al., 2025; Black et al., 2024; 2025). Leveraging large-scale, multistage pretraining, these models translate visual observations and natural language instructions into low-level joint-space commands, unifying multimodal perception with motor control. Although still a relatively new paradigm, current VLA systems already demonstrate previously unattainable generalization across environments, tasks, and robot embodiments. Since VLAs are designed for closed-loop interaction with the physical world, knowing when and how strongly to trust their actions is critical. For example, consider a robot performing a task in a safety-critical environment, or attempting to manipulate a valuable, fragile object. If the policy can accurately express uncertainty, then costly or dangerous accidents can be avoided, such as by refining the instruction or deferring the task to a human. Despite the importance of calibrated confidence, relevant questions about VLAs remain largely unaddressed, e.g., whether VLAs are calibrated, how calibration evolves over the task horizon, and how it can be improved.

**Contributions.** To bridge this gap, we present (to our knowledge) the first study of calibration in VLAs, identifying key open questions and introducing practical remedies for miscalibration. Our contributions include: **(1)** We perform a first-of-its-kind evaluation to measure the relationship between task success and calibration error across multiple datasets and VLA variants, finding that the model architecture and training objective may play significant roles in determining this relationship. **(2)** We propose a lightweight, Bayesian-style method that averages a VLA's confidence across multiple semantically equivalent rephrasings of an instruction. This approach consistently improves calibration, cutting expected calibration error by more than 20% on average. **(3)** We analyze calibration over task time, showing that confidence is often most reliable after making some task progress, suggesting natural points for risk-aware intervention. **(4)** We discover unpredictable over-/underconfidence in

Figure 1: To be trustworthy, a robotic system must be able to reliably express its confidence in its ability to perform a task, especially in high-stakes and open-world domains. A well-calibrated robot policy produces confidence estimates that align with its probability of task success. For example, the robot should succeed on 95% of instances for which it expresses 95% confidence.

different action dimensions and propose a method to recalibrate each action dimension independently to produce more reliable confidence estimates. Overall, our aim in this study is to begin to develop the tools and conceptual understanding necessary to render VLAs not only highly performant but also highly trustworthy via reliable uncertainty quantification. We discuss particular directions for future work in Section 5 , including applications to other architectures, active learning, and robot safety.

## 2 CALIBRATION IN VISION-LANGUAGE-ACTION MODELS

In this section, we formalize the problem of confidence calibration for vision-language-action models. We define calibration, describe a typical example of how contemporary VLA architectures generate actions and how to extract a corresponding confidence estimate, and introduce the standard calibration metrics used in our empirical study (see Figure 2 for a visual summary of this material). Finally, we present two lightweight remedies for the miscalibration we observe in practice: *prompt ensembles* and *action-wise Platt scaling*. A further discussion of related work on confidence calibration, VLAs, LLMs, and uncertainty quantification in robotics can be found in Appendix A.

**Calibration** Let $C \in [0, 1]$ denote the confidence reported by a robot policy and $Y \in \{0, 1\}$ the binary indicator of task success (we use uppercase $C, Y$ for random variables and lowercase $c, y$ for their realizations). A perfectly calibrated predictor (Guo et al., 2017) satisfies

$$\mathbb{P}(Y = 1 \mid C = c) = c, \qquad \forall c \in [0, 1]. \tag{1}$$

If this condition is met, then for the subset of trials on which the robot reports 80% confidence, we should observe successful completion 80% of the time.

**Vision-Language-Action Models** At task timestep $t$, a VLA (Kim et al., 2025; Zitkovich et al., 2023; Black et al., 2024) policy $\pi_\theta$ has access to the observed history $o_t = (v_{\leq t}, l_{\leq t}, a_{<t})$, where $v$ is a visual observation, $l$ is the (possibly fixed) natural language instruction, and $a$ is an action. The policy induces a distribution over the next action $\pi_\theta(a \mid o_t)$. When performing a task, the policy executes the most likely action $a_t^\star = \arg\max_a \pi_\theta(a \mid o_t)$. Given the need for calibration, we extract a scalar confidence score $c_t$ from the policy. We interpret $c_t$ as the policy's estimate of the probability that the task will ultimately succeed given the observation history and the chosen action, i.e., $\mathbb{P}(Y = 1 \mid o_t, a_t^\star)$.

**Baseline Confidence Estimation** Many state-of-the-art VLAs, including OpenVLA (Kim et al., 2025) and RT-2 (Zitkovich et al., 2023), represent an action with $D$ discrete tokens, where each represents one dimension of the robot's action space. For each dimension $d \in \{1, \ldots, D\}$ the policy outputs logits $z_t^{(d)}$ and probabilities $p_t^{(d)} = \text{softmax}(z_t^{(d)})$, $p_{t,k}^{(d)} = \pi_\theta(A_t^{(d)} = k \mid o_t)$ for action tokens $k \in \{1, \ldots, K\}$ in an action vocabulary of size $K$. At time $t$, the policy selects the top tokens $a_t^{(d)} = \arg\max_k p_{t,k}^{(d)}$ for each action dimension $d$, and decodes them into a continuous action for execution by the robot. Because VLAs with token-based decoders closely mirror LLMs, we can adopt the usual LLM heuristic of using the probability of the token actually chosen as a confidence signal (Kuhn et al., 2023). For each action dimension, we take the probability assigned to the selected

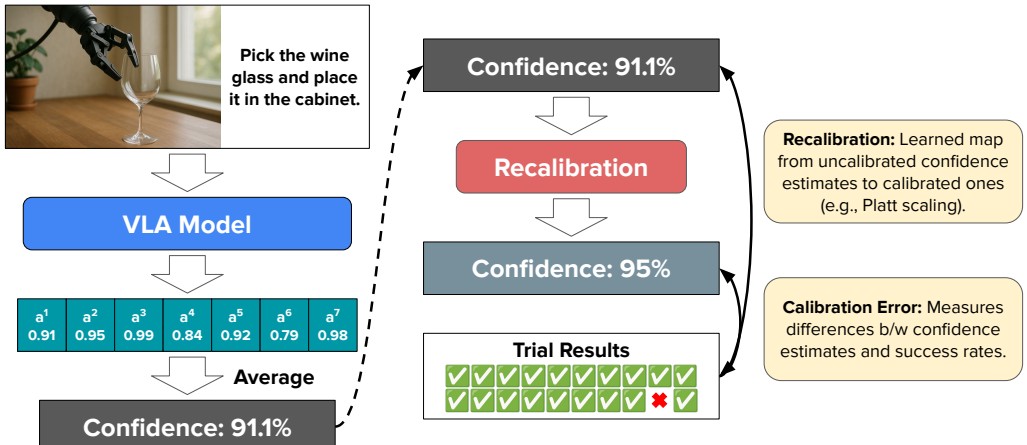

Figure 2: Given an input image and text instruction, popular VLAs such as OpenVLA and RT-2 generate a distribution over discrete action tokens for each of the robot's degrees of freedom. Confidence in each dimension's prediction can be estimated using the probability assigned to the predicted token; a single estimate can be produced by averaging across dimensions. Given an uncalibrated confidence estimate, recalibration methods such as Platt scaling use a small calibration dataset to learn a map from uncalibrated confidence estimates to calibrated ones. Calibration error can be measured by comparing confidence estimates to actual task success rates.

token, and average these values:

$$c_t \;=\; \frac{1}{D} \sum_{d=1}^{D} \max_k p_{t,k}^{(d)}. \tag{2}$$

Averaging plays the same role as length normalization in LLMs, preventing confidence from being unfairly reduced for robots with many degrees of freedom. We use this as our baseline confidence estimate in our experiments. The rest of this section treats $c_t$ as a black-box number, and so applies equally to, e.g., diffusion-based or other controllers, as long as they can emit such a scalar.

**Measuring Calibration**   Calibration metrics translate deviations from the condition in equation 1 into quantitative measures of *miscalibration*. Given the difficulty of measuring such a condition (i.e., comparing two distributions), we consider a range of metrics. This guards against corner cases and ensures that improvements in controlled evaluations translate into more actionable and reliable confidence estimates that are simultaneously calibrated, discriminative, and informative.

For what follows, let $\{(c_i, y_i)\}_{i=1}^N$ denote the reported confidence and binary outcome for each of $N$ robot trials (episodes). Each trial $i$ consists of timesteps $t = 1, \ldots, T_i$ with per-timestep confidences $c_{i,t}$; to obtain a single trial-level value $c_i$ we apply an aggregation function $h$: $c_i = h(\{c_{i,t}\}_{t=1}^{T_i})$. Possible choices include the confidence before the first action, the mean across timesteps, or the min/max over the trajectory. The following measures are agnostic to choice of $h$; in our experiments we use the pre-action confidence $c_i = c_{i,1}$, reflecting the high-stakes open-world robotics setting in which early risk assessment is particularly valuable.

One popular measure of miscalibration is **expected calibration error (ECE)** (Guo et al., 2017; Fisch et al., 2022):

$$\mathrm{ECE}_q \;=\; \left( \mathbb{E}_C \big[ |\mathbb{P}(Y = 1 \mid C) - C|^q \big] \right)^{1/q}. \tag{3}$$

ECE measures the expected difference between confidence and accuracy over the robot's task data distribution. The parameter $q$ is typically set to $q \in \{1, 2\}$, where $ECE_1$ weights deviations linearly and $ECE_2$ penalizes larger errors. Because we can observe only a finite sample of trial results, the conditional expectation in equation 3 cannot be directly measured. Instead, it is typically approximated with a binning-based estimator. With results from $N$ robot trials, we approximate the population ECE quantity in equation 3 by first ordering predictions according to confidence, and

splitting them into $M$ equal-sized bins $B_1, \ldots, B_M$. Then, our empirical estimate $\widehat{\text{ECE}}_q$ is given by

$$\widehat{\text{ECE}}_q \;=\; \Big(\sum_{m=1}^{M} \frac{|B_m|}{N} \big|\text{acc}(B_m) - \text{conf}(B_m)\big|^q\Big)^{1/q},$$

where $\text{acc}(B_m) = \frac{1}{|B_m|}\sum_{i \in B_m} y_i$ and $\text{conf}(B_m) = \frac{1}{|B_m|}\sum_{i \in B_m} c_i$. We report both $\text{ECE}_1$ and $\text{ECE}_2$ in our experiments.

Beyond ECE, a classic measure of the quality of probabilistic forecasting is **Brier score** (Brier, 1950): $\text{BS} = \frac{1}{N}\sum_{i=1}^{N}(c_i - y_i)^2$. Brier score is an example of a *proper scoring rule* (Gneiting & Raftery, 2007), meaning that it is minimized only when the predicted distribution matches the true distribution for each example (as opposed to ECE, where data is grouped by confidence level). Brier score rewards both reliability (confidence matching success) and sharpness (predictions away from the population rate). Another metric used to measure calibration (Guo et al., 2017) is **negative log-likelihood (NLL)**: $\text{NLL} = -\frac{1}{N}\sum_{i=1}^{N}\big[y_i \log c_i + (1 - y_i)\log(1 - c_i)\big]$. Also a proper scoring rule, NLL penalizes very confident failures much more heavily than Brier score.

**Prompt Ensembles** Similar to the VLMs from which they are derived (Zhou et al., 2024), semantically meaningless lexical differences in instructions, e.g., *"pick up the coffee cup"* vs. *"grab the mug"*, might shift visual attention, alter path planning, and thereby change the predictions and confidence scores emitted by VLA models. Although such sensitivity can pose a challenge to reliable task execution, it also creates an opportunity to employ an ensemble-based approach (Lakshminarayanan et al., 2017) to confidence estimation. Specifically, we can treat the particular wording of an instruction as a latent random variable and marginalize over it via Bayesian model averaging. Such averaging over rephrasings reduces variance in the final confidence estimate by canceling out the noise induced by word choice. Implementing this idea as an algorithm, we can: **(1) Generate rephrasings.** An auxiliary LLM produces $r$ semantically equivalent prompts $\mathcal{L}_{alt} = \{l_{alt}^{(1)}, \ldots, l_{alt}^{(r)}\}$ (see Table 2 for example). **(2) Estimate confidence with each variant.** Generating an action with $l_{alt}^{(i)}$ yields a confidence $c_t^{(i)}$. via equation 2. **(3) Aggregate.** The final estimate is the ensemble mean $c_t^{ens} = \frac{1}{r}\sum_{i=1}^{r} c_t^{(i)}$. Conceptually, this prompt ensemble technique can play a similar role to full model ensembles (Lakshminarayanan et al., 2017) or inference-time dropout (Gal & Ghahramani, 2016). With batched inference, this method should incur negligible latency with added inference time complexity $\mathcal{O}(1)$.

**Recalibration and Platt Scaling** A standard remedy for miscalibration in classification models is to gather a validation set from the task distribution and perform post hoc recalibration, learning a function that maps uncalibrated confidence estimates to calibrated ones (Platt, 1999; Naeini et al., 2015; Zadrozny & Elkan, 2001; Guo et al., 2017). Post hoc recalibrators are typically simple functions of the original confidence estimate. One popular example of a post hoc recalibration method is Platt scaling (Platt, 1999; Kumar et al., 2019). Given confidence outputs and binary outcomes $\{(c_i, y_i)\}_{i=1}^{N}$ on a held-out validation set, we can fit a transform $g(c) = \sigma(\alpha c + \beta)$ that minimizes NLL,

$$\min_{\alpha,\beta} -\sum_i \Big[y_i \log g(c_i) + (1 - y_i)\log\big(1 - g(c_i)\big)\Big],$$

where $\sigma(x) = 1/(1 + e^{-x})$. At inference each $c_t$ is replaced with $\tilde{c}_t = g(c_t)$, with the goal of aligning confidence with the probability of task success while leaving the model's actions unchanged. Unlike classification models that emit one predictive distribution per sample, token-based VLAs output one predictive distribution per action dimension. Those dimensions can differ dramatically, for example because gripper open/close appears in nearly 100% of demonstrations, whereas "rotate wrist" is rarer. A single global transform therefore may not be able to correct all dimensions simultaneously.

**Action-Wise Platt Scaling** We propose to address this heterogeneity by fitting one affine transform per dimension $d$,

$$\tilde{c}_t \;=\; \frac{1}{D}\sum_{d=1}^{D} \sigma\big(\alpha_d \max_k p_{t,k}^{(d)} + \beta_d\big),$$

with parameters $\{\alpha_d, \beta_d\}$ learned on the same calibration set. Intuitively, $\alpha_d$ scales dimension $d$, flattening or sharpening its distribution, while $\beta_d$ shifts its overall optimism. As in standard Platt

scaling, the chosen tokens $a_t^{(d)}$ remain unchanged; we only modify the reported confidence. This dimension-wise perspective can be applied to other post hoc recalibrators as well, for instance action-wise temperature scaling. Beyond the specific methodology, we aim to highlight that VLA calibration will require domain-specific tools rather than direct transplant of methods designed for other model classes.

## 3 EXPERIMENTS

Having described the problem of confidence calibration in vision-language-action models, we next conduct an empirical investigation, studying the following questions: **(1)** Does a higher success rate imply better calibration (Section 3.1)? **(2)** Can prompt ensembles consistently improve confidence estimates (Section 3.2)? **(3)** How does calibration evolve over the task time horizon (Section 3.3)? **(4)** Are action dimensions differentially calibrated, and what are the implications for recalibration (Section 3.4)? Our aim is to highlight key issues and lay the empirical groundwork for future research on calibrating VLAs.

We perform our experiments using 4 different VLA variants: OpenVLA, MolmoAct (Lee et al., 2025), UniVLA (Bu et al., 2025), and NORA (Hung et al., 2025). We choose these models because they each predict tokens in a manner that allows for a natural probabilistic interpretation.[1] All models are fine-tuned on 4 different task suites from the LIBERO (Liu et al., 2023) benchmark: Spatial, Object, Goal, and 10. LIBERO is a simulation environment for language-conditioned robot manipulation tasks inspired by human activities (see Table 3 for task examples). We also include results for the 8-bit and 4-bit quantized versions of OpenVLA fine-tuned on Spatial, Object, and Goal, for a total of **22 model/task suite combinations**. Each task suite features 10 different tasks with 50 randomized initializations, for a total of 500 examples. For a given task suite, task success rate represents the proportion of the 500 trials that result in success (while task error rate represents the proportion of trials that result in failure). To calculate ECE, we use 12 equal-mass bins and the Python package released with Kumar et al. (2019). Unless otherwise noted, we focus on confidence estimates produced at the first timestep, before any action is taken. This aligns with the need for safety in open-world robot deployments, where robots should signal uncertainty as early as possible in order to avoid costly or dangerous incidents. In Section 3.3, we explore how calibration differs across timesteps over the task horizon.

### 3.1 RELATIONSHIP BETWEEN CALIBRATION AND TASK SUCCESS

Much of the early research on calibration in deep learning models was based on perceptions of how task success (usually image classification accuracy) was related to calibration error. In particular, influential work argued that modern neural networks are poorly calibrated (Guo et al., 2017), suggesting that improved accuracy comes at the expense of calibration. This led researchers to propose training interventions to augment the cross-entropy loss with other objectives seeking improved calibration, potentially at a cost to overall accuracy (Kumar et al., 2018; Mukhoti et al., 2020). However, subsequent research found that, in fact, more accurate networks are generally better calibrated (and easier to recalibrate), and thus modifications to training procedures may not be needed (Minderer et al., 2021). Instead, techniques such as post hoc recalibration (Guo et al., 2017; Zadrozny & Elkan, 2001; Kumar et al., 2019) and ensembling (Lakshminarayanan et al., 2017; Fort & Lakshminarayanan, 2024), applied to models trained for high accuracy, are sufficient to achieve low calibration error (although calibration under distribution shift remains a significant challenge (Ovadia et al., 2019)).

To establish high-level direction for calibration research in VLAs, our first experiment focuses on this important question of how task success relates to calibration error (according to $ECE_1$, $ECE_2$, Brier score, and NLL) across the 22 model/task suite combinations described above. Results are visualized in Figure 3, and also reported in Table 5. All models exhibit a roughly monotone relationship between task error and the discriminative metrics (Brier score and NLL), but they differ on ECE: OpenVLA and MolmoAct tend to achieve lower ECE when task error is low, whereas UniVLA and NORA do not show as clean a trend. One possible explanation is that in OpenVLA and MolmoAct, the cross-entropy loss (a proper scoring rule) more directly supervises discretized action dimension tokens, so bin-averaged confidence better tracks per-task success. By contrast, the latent or compressed action representations and auxiliary objectives in UniVLA and NORA may introduce a more significant

---

[1]Descriptions of confidence estimation with MolmoAct, UniVLA, and NORA are found in Appendix C.1.

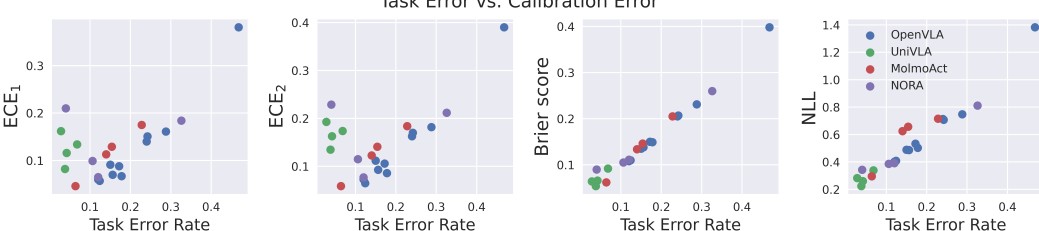

Figure 3: Visualization of task error rates compared against 4 different calibration error measurements for 4 VLA variants (OpenVLA, MolmoAct, UniVLA, and NORA) and 4 LIBERO task suites (Spatial, Object, Goal, 10), as well as OpenVLA 8- and 4-bit versions on Spatial, Object, and Goal. All models exhibit a roughly monotonic relationship between task error and the discriminative measures (Brier score and NLL). ECE shows differences between models, potentially due to architecture and objective differences.

mismatch between token probabilities and success. These patterns are only suggestive, but they point to architectural complexity in modern VLAs as a potential source of unfavorable calibration behavior. Future work with controlled ablations is needed to fully understand which design choices drive the observed behavior.

### 3.2 ENSEMBLING CONFIDENCE OVER PROMPTS

We next study the empirical effectiveness of the prompt ensemble approach described in Section 2. First, for the natural language instruction associated with each task, we create 20 rephrasings using GPT-4o-mini (see Appendix Table 4 for prompts used). During testing, we produce a confidence estimate conditioned on each "reprompt", and average over these confidence scores to obtain the final model confidence. We evaluate OpenVLA and its quantized variants across 3 tasks suites, as well as UniVLA and NORA on the Spatial suite. We measure $ECE_1$, $ECE_2$, Brier score, and NLL, comparing to the baseline method for producing confidence estimates using the original instruction.

Detailed results are recorded in Table 1 (results for the 8-bit and 4-bit quantized model are deferred to Appendix Table 6). Across all models and task suites, the Reprompt (prompt ensemble) approach always decreases calibration error according to $ECE_1$, $ECE_2$, and NLL, and never performs worse according to any of the metrics. Decreases in ECE can reach up to 40%, and average decreases are more than 20% (see Figure 7 for a visualization of percent changes across all metrics). The improvement is larger for ECE than the proper scoring rules (Brier score and NLL), where average miscalibration decreases are closer to 5-10%. This suggests that improvements in reliability (i.e., confidence matching marginal success rates) are generally greater than those in sharpness, which is expected from an ensemble technique targeted at reducing variance in confidence estimation. Overall, given its lightweight nature and strong empirical effectiveness, such data augmentation approaches seem like a promising direction for enhancing uncertainty quantification in VLAs.

To understand the robustness of these results, we perform multiple ablations; because of space constraints, we defer full details and results to Appendix D.3. First, we consider the effect of changing the prompt given to GPT-4o-mini for producing the 20 instruction rephrasings, finding that improvements in calibration error from the Reprompt method are robust to different rephrasing prompts. Second, we consider the effect of the number of prompts used in the prompt ensemble, and see that the algorithm behaves favorably, where calibration error generally decreases as more instructions are included in the ensemble.

### 3.3 CALIBRATION OVER TASK TIME

The previous experiments evaluate confidence before the first action is executed, a conservative choice for safety-critical deployments. Yet many tasks might allow the robot to collect more information without risk before having to express confidence. For instance, in the wine glass scenario of Figure 1, the gripper can hover above the stem, refine its scene representation, and only then decide whether it is confident enough to proceed. More context should, in principle, yield better calibration.

To test this intuition we measure calibration across 500 test trials for 100 different levels of task completion ($\{0, 1, \ldots, 99\}\%$), where task completion is calculated as the current timestep index $t$ divided by the total number of timesteps in the task episode (and multiplied by 100). Results for each

| Model | Dataset | Method | $ECE_1$ | $ECE_2$ | Brier | NLL |
|-------|---------|--------|---------|---------|-------|-----|
| OpenVLA | Spatial | Baseline | 0.088 | 0.106 | 0.150 | 0.533 |
| | Spatial | Reprompt | **0.052** | **0.068** | **0.145** | **0.477** |
| | Object | Baseline | 0.060 | 0.073 | 0.108 | 0.401 |
| | Object | Reprompt | **0.036** | **0.053** | **0.105** | **0.361** |
| | Goal | Baseline | 0.151 | 0.170 | 0.207 | 0.707 |
| | Goal | Reprompt | **0.115** | **0.132** | **0.197** | **0.620** |
| UniVLA | Spatial | Baseline | 0.162 | 0.193 | 0.064 | 0.282 |
| | Spatial | Reprompt | **0.157** | **0.192** | 0.064 | **0.262** |
| NORA | Spatial | Baseline | 0.099 | 0.115 | 0.105 | 0.386 |
| | Spatial | Reprompt | **0.092** | **0.112** | 0.105 | **0.376** |

Table 1: Calibration error measurements for 2 different methods of confidence estimation: (1) baseline (average selected token probability); (2) Reprompt (ensembling over semantically equivalent prompts). Prompt ensembling consistently improves all measures, and never leads to worse calibration.

level of task completion are averaged across the 500 trials, to study whether high-level confidence and calibration trends might occur. We focus on 6 model/task suite combinations (OpenVLA and its 8-bit version, each applied to Spatial, Object, and Goal suites), to understand how any observations might generalize. Our goal is to examine whether the quality of confidence estimates changes as the robot progresses in its task. At each measured timestep, we compute $ECE_1$ and Brier score using the baseline method for producing confidence estimates. Because a downstream safety monitor might consider the history of recent estimates (beyond the current one), we evaluate three aggregation rules for reporting a confidence estimate at each timestep: **Current** - confidence from the current timestep only; **Window (5)** - mean over the baseline confidence estimates from the current timestep and the four immediately preceding ones; **Avg. All** - mean over the baseline confidence estimates from all steps seen so far.

Beyond calibration error, we also visualize how average confidence estimates evolve over the task horizon, separated by successful and failed trials. Finally, we plot reliability diagrams for the timesteps corresponding to $\{0, 50, 75, 99\}\%$ completion. Reliability diagrams (Guo et al., 2017) offer a visualization of expected calibration error, using a similar binning strategy. They show confidence against accuracy for each bin, where a well-calibrated system lies on the $x = y$ line. Results for the Spatial task suite with the fine-tuned OpenVLA model are shown in Figure 4. Additional results for Spatial with the Quant-8 model (Figure 11), as well as the Object and Goal suites with the full precision (Figures 12, 14) and Quant-8 models (Figures 13, 15), are presented in Appendix D.4.

Focusing first on results using the confidence estimate from the current step, a clear pattern emerges. As shown in Figure 4 and Figures 11-15, across all 6 settings and both metrics calibration improves sharply from 0% to approximately 50% completion, then plateaus or deteriorates back towards the original level (left-most and left-middle plots in section (a) of the results figures). Beyond lower scores on calibration error, our additional plots characterize the improved probabilistic nature of the confidence estimates towards the middle of the task horizon. In the right-middle and right-most plots in top section (a) (titled "Successful Trials" and "Failed Trials"), the gap between the average confidence on successful vs. failed trials tends to be greatest around 50% task completion. Additionally, considering the top row of reliability diagrams (marked (b)), we can see that the confidence estimates using the current step become far more reliable around this point: error for many bins is large at the beginning, then the difference between confidence and accuracy for most bins becomes smaller until the task is roughly 75% complete. Near task completion, the gap between confidence on successful vs. failed trials closes again, and reliability suffers as a result.

These results support our hypothesis regarding the utility of gathering more information before expressing a confidence estimate, even using the simple baseline method. Early miscalibration might be expected: with only a high-level image frame and a textual instruction, the policy lacks much information about how hard the task might be. By mid-trajectory, it might observe how its gripper and the scene interact, analogous to a "test grip" on the wine glass, so its probability estimates may better match reality. Near the goal state, however, calibration error is generally worse, for instance because of differences in the action dimensions being emphasized in each phase of the task. These experiments suggest a practical recipe: let the robot execute a certified-safe prefix of the trajectory,

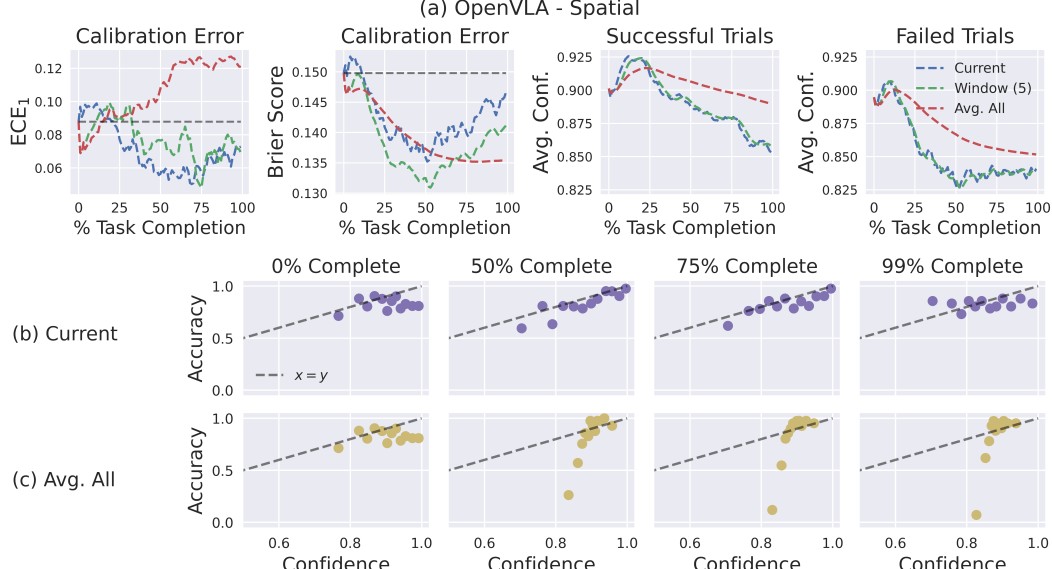

Figure 4: Empirical study of calibration error across the task time. In the top row (a), the left two plots show how calibration evolves with task progress, while the right two plots show the average confidence by task time, grouped by successful and failed trials. The bottom two rows (b, c) offer a sample of the reliability diagrams produced by different methods for aggregating confidence estimates across time. Overall, these results illustrate that calibration can improve as the task progresses and more information is gathered, suggesting opportunities for context-aware uncertainty interventions.

and then assess its confidence and intervene if necessary. Such horizon-aware monitoring balances safety (no contact made during the prefix) with the improved reliability in confidence estimates that comes from extra perceptual context.

We next observe the effects of other simple strategies for aggregating across timesteps. Specifically, we consider a sliding window average over 5 single-step estimates, and the average of all observed single-step estimates. Across all model/task suite combinations, the sliding window method is particularly effective at improving the Brier score around the middle of the task horizon, although ECE is not always improved. With respect to averaging all single-step estimates so far, we observe that Brier score improves throughout the task, but ECE generally gets worse. The underlying behavior driving these changes can be observed in the bottom row of reliability diagrams (marked (c)): the "Avg. All" method is fairly successful at sorting the failed examples to have relatively lower confidence, but also reduces variance in the confidence estimates such that they are highly overconfident in this range. Thus, while this approach fares poorly according to ECE, it is actually promising in the sense that it enables more effective discrimination between successful and failed trials.

Finally, to understand the generality of these results, we repeat these experiments using the UniVLA model, presented in Appendix Figures 16, 17, and 18. We once again find that calibration improves after making some task progress before deteriorating again, suggesting opportunities for context- and risk-aware applications of confidence quantification.

**Qualitative Examples of Context-Aware Confidence Monitoring**    To illustrate how context-aware confidence monitoring could work in practice, we apply it to a representative pick and place task from the Goal suite. The task is to "put the wine bottle on the rack", a case where the robot should be relatively conservative to avoid breaking glass. We consider a naive approach to context-aware monitoring, proposing to halt task performance when both: **(1)** the confidence level falls below a threshold set to the 10% quantile of confidence estimates for that point in the task horizon across all task trials (based on percent completion); **(2)** the robot is within a few inches of contacting an object, or already has contacted an object. Since building a system to detect proximity to objects is beyond the scope of this work, we focus on a qualitative demonstration of this idea. For each example under examination, we plot current confidence for $\{0, 20, 40, 60, 80, 98\}\%$ completion, as well as the corresponding 10% quantile risk threshold and an image of the robot environment at that time.

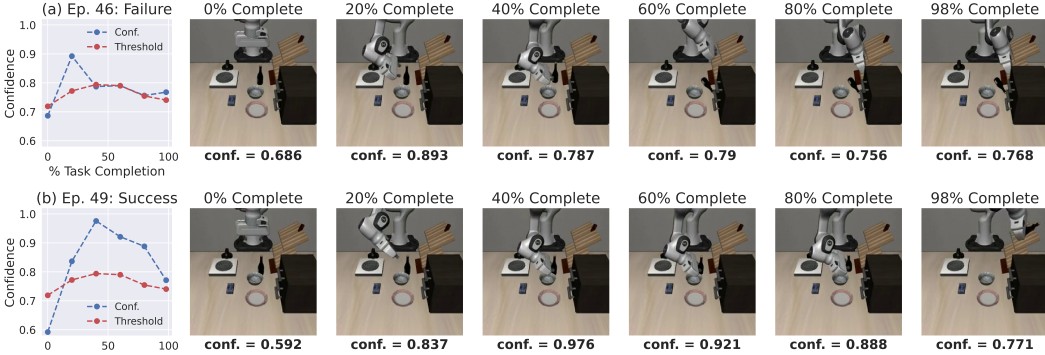

Figure 5: Qualitative examples of a context-aware confidence monitoring strategy applied to a task from the Goal suite. Here, the task is to "put the wine bottle on the rack". The red dashed line represents the 10% quantile of the confidence estimates output by the model across the task time horizon, offering a potential threshold below which the robot may abstain from performing the task.

Some illustrative examples are shown in Figure 5. In both Episode 46 (marked (a)) and Episode 49 (marked (b)), confidence begins below the threshold. However, given the knowledge that calibration improves throughout the task horizon, we may prefer to allow the robot to proceed with the task as long as it is not too near any objects. Comparing these examples shows the potential of such an approach. In Episode 46, although confidence increases throughout the beginning of the task, it dives sharply after 40% task completion and remains low throughout the rest of the task, as the wine bottle is ultimately knocked down and the model is unable to recover. Given our proposed strategy, the task might have been halted before the wine bottle was contacted (at 40% completion), and could, for example, have been deferred to a human. On the other hand, in Episode 49 confidence estimates remain above the threshold for the rest of the trial, and the robot is successful. This episode also highlights the usefulness of an adaptive confidence threshold, given that the confidence estimate at 98% task completion would have fallen below the threshold at other timesteps. More such qualitative examples are provided in Appendix Figure 19, including another detailed case analysis. These include both cases where the strategy succeeds and others where it fails (e.g., in Episode 1 confidence falls below the threshold while grasping in an ultimately successful trial).

### 3.4 Calibration Across Action Dimensions

Token-based VLAs such as OpenVLA or MolmoAct decompose a low-level command into tokens corresponding to 7 different action dimensions, predicted sequentially by an image-conditioned autoregressive LLM. Our baseline confidence estimate collapses this structure into a single scalar by averaging the top-token probabilities across the dimensions; implicitly, this assumes that every dimension is calibrated to the same degree. That assumption may not hold in practice: a gripper 'open' or 'close' token may appear in nearly every demonstration, whereas a $90°$ wrist roll could be rare. Such dataset imbalances and other implementation details might skew calibration across dimensions, and relevant algorithms, e.g., for post hoc recalibration, might benefit from explicitly addressing this. To study this question, our final experiment consists of two parts. **(1) Per-dimension calibration audit.** For each dimension $d$ we treat $\max_k p_{t,k}^{(d)}$ as that dimension's confidence and compute $\text{ECE}_1$, to examine whether different degrees of freedom are differentially calibrated. **(2) Targeted recalibration test.** We compare classic Platt scaling to the action-wise variant of Section 2, which learns independent scaling parameters for each action dimension before averaging the transformed confidences across dimensions.

We perform our study using MolmoAct and all 3 OpenVLA variants (full, Quant-8, Quant-4) fine-tuned on the Spatial and Goal task suites. Each experiment is run for 1000 trials, with random 20%/80% calibration-test splits. Given the reduced size of the test set, we measure calibration with 10 equal-mass bins (instead of 12 as in other experiments). Beyond traditional Platt scaling, we also include a range of baselines: temperature scaling (Guo et al., 2017), histogram binning (Zadrozny & Elkan, 2001), and Platt binning (Kumar et al., 2019). (Note: temperature scaling is excluded from MolmoAct results, as the variable-length action encodings preclude a straightforward adaptation.) Figure 6 shows results with the baseline estimates for the full precision OpenVLA and MolmoAct models; Appendix Figure 20 includes the results with the quantized models.

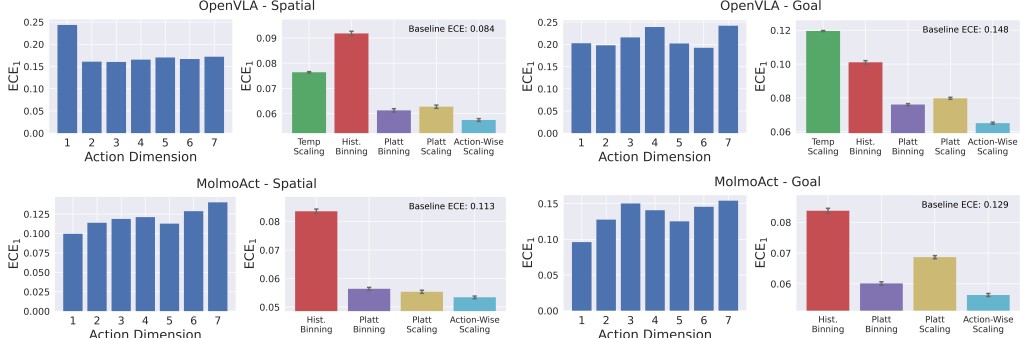

Figure 6: On the left of each pair of plots, we compare miscalibration across action dimensions. On the right, we compare the performance of baselines to action-wise Platt scaling.

Two key observations arise. First, $ECE_1$ varies by up to 2 times across dimensions, with no consistent best or worst dimensions across settings. It follows that a single scalar confidence masks significant differences in dimension-wise confidence estimates. Second, we can observe that replacing global Platt scaling with per-action-dimension transforms consistently lowers calibration error (and also improves over the additional baselines). ECE is improved by over 20% in some cases, without changing the selected tokens or adding meaningful runtime overhead. Together, these results indicate that VLA calibration research should treat each degree of freedom as its own concern in order to improve effectiveness of relevant algorithmic tools. Dimension-aware post hoc methods such as action-wise Platt scaling offer a simple yet effective path toward that goal. More broadly, our findings underscore that calibrating VLAs introduces challenges not seen in other domains, so meaningful progress will demand substantial domain-specific research rather than wholesale adoption of understanding and techniques from other areas of deep learning.

## 4 LIMITATIONS

All experiments in this work are conducted in simulation. This choice is common in robotics research (e.g., Chua et al. (2018); Janner et al. (2019); Tamar et al. (2016)), and enables the controlled resets and hundreds of rollouts per task suite needed for stable calibration estimates. However, it limits domain diversity and leaves open questions of how these results translate to real settings. Replicating the same study on physical robot hardware would require thousands of physical executions, which can be prohibitively time-consuming and expensive. Accordingly, simulation remains the most practical testbed for calibration research in the near term, though it cannot fully capture sensor noise, latency, wear-and-tear, and other physical factors that may influence confidence calibration in the real world. However, studying these techniques on real robots will ultimately be essential for understanding how these and other factors affect VLA calibration under realistic conditions.

## 5 FUTURE WORK

Several extensions follow naturally from our work. Confidence calibration should be benchmarked across other environments, robot embodiments, and VLA architectures, including diffusion-based planners, continuous regression heads, and flow-matching controllers. Establishing strong baselines for these architectures will likely require new confidence surrogates and post hoc adjustments. Likewise, perturbation techniques besides instruction paraphrasing, e.g., inserting synthetic lighting changes or random distractors, could expose additional failure modes and inspire new multimodal ensemble techniques. Running such experiments on real robots will be essential for understanding how various physical factors affect VLA calibration in the real world. Uncertainty estimation also holds the potential to guide efficient data gathering and fine-tuning, as in active learning (Wang et al., 2017). Further, while all of our experiments focused on fine-tuned in-distribution settings, future work should consider these phenomena in the zero-shot and out of distribution settings. Finally, our temporal analysis hints at a period mid-trajectory where confidence is most trustworthy; integrating that signal into selective execution, planning, or human-in-the-loop systems is an open challenge for designing adaptive risk-aware robotic pipelines.

## REPRODUCIBILITY STATEMENT

To ensure reproducibility, all of our code will be shared upon publication of this paper. All of the models and datasets we use are available for public download and use.

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

# A  RELATED WORK

## A.1  CONFIDENCE CALIBRATION

A model is considered well-calibrated when the confidence (i.e., probability) it assigns to an outcome matches the long-run frequency of that outcome. Deviation from this condition is often quantified via expected calibration error (ECE) (Guo et al., 2017), which is approximated by grouping predictions with similar confidence into bins and measuring the absolute difference between confidence and accuracy in each bin. Beyond ECE, other metrics such as maximum calibration error (Guo et al., 2017), Brier score (Brier, 1950), and negative log-likelihood (NLL) are used to capture related and complementary notions of miscalibration. Despite their impressive accuracy and trends toward better calibration, modern neural networks still display some persistent miscalibration (Minderer et al., 2021), especially when the task distribution does not match the training distribution (Ovadia et al., 2019). Consequently, a rich toolbox of post hoc fixes has emerged: Platt scaling (Platt, 1999), temperature scaling (Guo et al., 2017), histogram binning (Zadrozny & Elkan, 2001), and other recalibration methods (Kumar et al., 2019; Naeini et al., 2015; Zollo et al., 2024) can be applied to an already trained model to shrink residual calibration error without altering its decision rule. Other popular methods to improve calibration of neural networks include dropout (Gal & Ghahramani, 2016) and ensembling (Lakshminarayanan et al., 2017).

### A.2 VISION-LANGUAGE-ACTION MODELS

*Vision-Language-Action* models take visual data and natural language instructions as input and output robot actions (Zitkovich et al., 2023; Kim et al., 2025; Black et al., 2024; 2025). They are typically initialized from a visually-conditioned language model (VLM) that, in turn, is initialized from a pretrained large language model (LLM). This allows VLAs to leverage rich multimodal priors in joining perception and action generation into a single end-to-end pipeline. While some VLAs retain the token-based output paradigm inherited from these base models (Zitkovich et al., 2023; Kim et al., 2025), others have augmented the architecture with, e.g., flow matching action experts for smooth, high-frequency control (Black et al., 2024; 2025). These systems already perform complex, language-specified tasks across diverse environments and robot embodiments. However, they lack a reliable mechanism for quantifying the uncertainty of their chosen action sequences.

### A.3 CALIBRATION IN LLMS

Given that most state-of-the-art VLAs are built on LLM backbones, it is natural to consider how our work relates to existing studies of calibration in LLMs, an area that has received considerable attention recently. Typical approaches to generating confidence estimates with LLMs include expressing calibration as a multiple-choice question for the LLM (Kadavath et al., 2022), directly verbalizing confidence with the model's text output (Lin et al., 2022; Tian et al., 2023; Band et al., 2024), or measuring semantic consistency across many sampled outputs for the same query (Kuhn et al., 2023; Duan et al., 2024; Chen et al., 2024). However, it is difficult to adapt these methods to the VLA setting. For example, while LLMs may produce the entire sequence before measuring confidence, in robotics potential failures must be flagged much earlier in the trajectory to ensure safety and avoid costly accidents. Also, sampling multiple full trajectories in the physical world may be impossible. Finally, current VLAs lack the flexible and robust text-to-text interface of LLMs, and thus cannot be expected to, e.g., answer natural language questions about their confidence in an action prediction.

### A.4 UNCERTAINTY QUANTIFICATION IN ROBOTICS

Uncertainty quantification has long been a focus in robotics, particularly through the lens of *probabilistic robotics* (Thrun et al., 2005; Deisenroth & Rasmussen, 2011). Recent work explores the use of model ensembles in control systems, examining how disagreements among multiple models can capture uncertainty and guide more cautious policy updates and actions (Chua et al., 2018; Pathak et al., 2019; Ataei & Dhiman, 2024). Other advances include uncertainty-based active exploration (Wang et al., 2024) and the application of conformal methods, which can provide distribution-free guarantees for planning or classification pipelines (Ren et al., 2023; Sun et al., 2023). How to obtain calibrated, task-level uncertainty estimates given the expressive, multimodal perception and control abilities of VLAs remains an open and safety-critical problem.

## B  LLM USAGE

OpenAI o3 was used to polish and proofread the writing in this paper.

| Instruction | Pick up the black bowl between the plate and the ramekin and place it on the plate |
|---|---|
| Rephrasings | (1) Lift the black bowl located between the plate and the ramekin and set it on the plate.
(2) Grasp the black bowl found between the plate and the ramekin and move it to the plate.
(3) Take the black bowl positioned between the plate and the ramekin and position it on the plate. |

Table 2: Example of multiple paraphrases of a single VLA instruction (from our experiments with LIBERO Spatial). The rephrasings carry the same meaning as the original instruction, enabling a *reprompting* approach in which we ensemble over predictions conditioned on lexically different yet semantically equivalent instructions.

| Dataset | Example Image | Example Instruction |
|---|---|---|
| Spatial |  | Pick up the black bowl between the plate and the ramekin and place it on the plate. |
| Object |  | Pick up the alphabet soup and place it in the basket. |
| Goal |  | Open the middle drawer of the cabinet. |
| 10 |  | Put both the alphabet soup and the tomato sauce in the basket. |

Table 3: Task examples from the LIBERO Spatial, Object, and Goal task suites.

## C   ADDITIONAL EXPERIMENT DETAILS

Task examples from the LIBERO Spatial, Object, and Goal task suites are shown in Table 3.

### C.1   CONFIDENCE ESTIMATION WITH VLA VARIANTS

**MolmoAct**   At each timestep, MolmoAct predicts a sequence of discrete action tokens for every action dimension, with a softmax distribution over the 256 action bins for each token. For a given

action dimension, we take the probability assigned to the executed bin for each of its tokens and average these probabilities to obtain a per-dimension confidence. This yields one scalar confidence value per action dimension. We then average these per-dimension confidences across all action dimensions to obtain a single scalar confidence for the action at that timestep.

**UniVLA**    At each timestep, UniVLA predicts a fixed number (4) of discrete latent action tokens, each chosen from a codebook of size 16. For each latent token, we take the softmax probability assigned to the selected code as its token-level confidence. Because these latent tokens are jointly decoded into all continuous control dimensions, the confidence scores do not correspond to particular action dimensions. Finally, we average these per-token confidences to obtain a single scalar confidence for UniVLA at that timestep.

**NORA**    At each timestep, NORA predicts an action chunk encoded as a sequence of discrete tokens, each selected via a softmax over the augmented action vocabulary. For every token, we take the softmax probability of the sampled token as its token-level confidence. Because the NORA tokenizer mixes information from all control dimensions, we do not derive per-dimension confidences. Instead, we compute a single scalar confidence for NORA by averaging these token-level confidences across all tokens in the predicted action chunk for that timestep.

## C.2    Rephrasing Prompts

The prompts given to GPT-4o-mini for rephrasing instructions from the LIBERO robot simulation environment are listed in Table 4. Prompt 1 is the main prompt used throughout the experiments, and Prompts 2 and 3 are used to ablate the sensitivity to the rephrasings. Note that the different prompts lead to substantially different rephrasings, as Prompts 2 and 3 mandate the retention of the words "pick" and "place", while as Table 2 shows, Prompt 1 leads to these words being replaced.

| | Prompt |
|---|---|
| Prompt 1 | You are generating alternative phrasings of a robotic task instruction while preserving its exact meaning.
### Task Instruction: '[TASK DESCRIPTION]'
### Instructions: - Generate **20** alternative ways to phrase the task instruction. - Keep each instruction **concise and unambiguous**. - Ensure the instructions remain suitable for a **robot, not a human**. - Only make **semantically meaningless** changes (e.g., word order, synonyms, slight rewording). - Double-check that the new instructions mean the same exact thing for the robot; do not just substitute synonyms without considering context. - Do **not** introduce additional steps, remove essential details, or alter the action.
### Output Format: Each rephrased instruction should be wrapped in '[instruction]' and '[/instruction]' tags, like this: [instruction] Rephrased instruction 1 [/instruction] [instruction] Rephrased instruction 2 [/instruction] |
| Prompt 2 | You are generating alternative phrasings of a robotic task instruction while preserving its exact meaning.
### Task Instruction: '[TASK DESCRIPTION]'
### Instructions: - Generate **20** alternative ways to phrase the task instruction. - Keep each instruction **concise and unambiguous**. - Ensure the instructions remain suitable for a **robot, not a human**. - Only make **semantically meaningless** changes (e.g., word order, synonyms, slight rewording). - Double-check that the new instructions mean the same exact thing for the robot; do not just substitute synonyms without considering context. - Do **not** introduce additional steps, remove essential details, or alter the action.
- The first word of the instruction should be 'PICK', and then it should also include the word 'PLACE'.
### Output Format: Each rephrased instruction should be wrapped in '[instruction]' and '[/instruction]' tags, like this: [instruction] Rephrased instruction 1 [/instruction] [instruction] Rephrased instruction 2 [/instruction] |
| Prompt 3 | You are generating alternative phrasings of a robotic task instruction while preserving its exact meaning.
### Task Instruction: '[TASK DESCRIPTION]'
### Instructions: - Generate **20** alternative ways to phrase the task instruction. - Keep each instruction **concise and unambiguous**. - Ensure the instructions remain suitable for a **robot, not a human**. - Only make **semantically meaningless** changes (e.g., word order, synonyms, slight rewording). - Double-check that the new instructions mean the same exact thing for the robot; do not just substitute synonyms without considering context. - Do **not** introduce additional steps, remove essential details, or alter the action. - Make the changes as minor as possible, as the robot's language system is not very robust to rephrasing. - The first word of the instruction should be 'PICK', and then it should also include the word 'PLACE'.
### Output Format: Each rephrased instruction should be wrapped in '[instruction]' and '[/instruction]' tags, like this: [instruction] Rephrased instruction 1 [/instruction] [instruction] Rephrased instruction 2 [/instruction] |

Table 4: Prompts 1, 2, and 3 for GPT-4o-mini to rephrase LIBERO task instructions. These prompts are used for Spatial and Object. There is a small modification for Goal.

| Dataset | Model | Quant | ECE$_1$ | ECE$_2$ | Brier | NLL | Succ. Rate |
|---------|-------|-------|---------|---------|-------|-----|------------|
| Spatial | NORA | - | 0.099 | 0.115 | 0.105 | 0.386 | 0.894 |
| Object | NORA | - | 0.210 | 0.229 | 0.090 | 0.343 | 0.960 |
| Goal | NORA | - | 0.065 | 0.077 | 0.111 | 0.391 | 0.880 |
| 10 | NORA | - | 0.184 | 0.212 | 0.260 | 0.811 | 0.674 |
| Spatial | MolmoAct | - | 0.113 | 0.123 | 0.134 | 0.625 | 0.860 |
| Object | MolmoAct | - | 0.046 | 0.059 | 0.062 | 0.296 | 0.936 |
| Goal | MolmoAct | - | 0.129 | 0.141 | 0.146 | 0.657 | 0.846 |
| 10 | MolmoAct | - | 0.175 | 0.184 | 0.205 | 0.716 | 0.772 |
| Spatial | UniVLA | - | 0.162 | 0.193 | 0.064 | 0.282 | 0.972 |
| Object | UniVLA | - | 0.082 | 0.135 | 0.054 | 0.224 | 0.962 |
| Goal | UniVLA | - | 0.116 | 0.163 | 0.066 | 0.261 | 0.958 |
| 10 | UniVLA | - | 0.134 | 0.174 | 0.092 | 0.339 | 0.932 |
| Spatial | OpenVLA | - | 0.088 | 0.106 | 0.150 | 0.533 | 0.828 |
| Object | OpenVLA | - | 0.060 | 0.073 | 0.108 | 0.401 | 0.880 |
| Goal | OpenVLA | - | 0.151 | 0.170 | 0.207 | 0.707 | 0.758 |
| 10 | OpenVLA | - | 0.381 | 0.390 | 0.398 | 1.382 | 0.532 |
| Spatial | OpenVLA | Quant-8 | 0.070 | 0.093 | 0.138 | 0.488 | 0.844 |
| Object | OpenVLA | Quant-8 | 0.057 | 0.065 | 0.110 | 0.409 | 0.876 |
| Goal | OpenVLA | Quant-8 | 0.140 | 0.163 | 0.205 | 0.713 | 0.760 |
| Spatial | OpenVLA | Quant-4 | 0.067 | 0.086 | 0.149 | 0.503 | 0.822 |
| Object | OpenVLA | Quant-4 | 0.091 | 0.112 | 0.135 | 0.489 | 0.850 |
| Goal | OpenVLA | Quant-4 | 0.161 | 0.182 | 0.231 | 0.748 | 0.712 |

Table 5: All task success and calibration results for 22 VLA/task suite combinations.

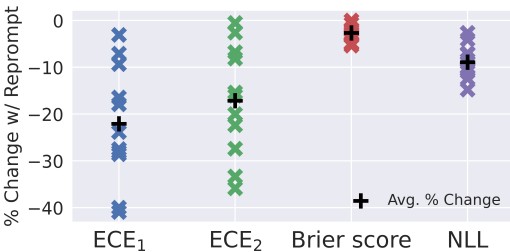

Figure 7: Percent change in calibration error measurements by ensembling over semantically equivalent prompts.

## D  ADDITIONAL EXPERIMENT RESULTS

All task success and calibration results for 22 VLA/task suite combinations are shown in Table 5.

### D.1  RELATIONSHIP BETWEEN TASK SUCCESS AND CALIBRATION

Table 5 features all results plotted in Figure 3.

### D.2  PROMPT ENSEMBLING

Figure 7 shows trends in percent change in calibration error using the prompt ensemble method.

### D.3  ABLATIONS FOR PROMPT ENSEMBLES

To understand the robustness of these results, we perform multiple ablations. First, we consider the effect of changing the prompt given to GPT-4o-mini for producing the 20 instruction rephrasings.

| Model | Dataset | Method | ECE$_1$ | ECE$_2$ | Brier score | NLL |
|---|---|---|---|---|---|---|
| OpenVLA Quant-8 | Spatial | Baseline | 0.070 | 0.093 | 0.138 | 0.488 |
|  |  | Reprompt | **0.050** | **0.062** | **0.131** | **0.434** |
|  | Object | Baseline | 0.057 | 0.065 | 0.110 | 0.409 |
|  |  | Reprompt | **0.041** | **0.052** | **0.108** | **0.375** |
|  | Goal | Baseline | 0.140 | 0.163 | 0.205 | 0.713 |
|  |  | Reprompt | **0.117** | **0.152** | **0.194** | **0.608** |
| OpenVLA Quant-4 | Spatial | Baseline | 0.067 | 0.086 | 0.149 | 0.503 |
|  |  | Reprompt | **0.055** | **0.079** | **0.148** | **0.482** |
|  | Object | Baseline | 0.091 | 0.112 | 0.135 | 0.489 |
|  |  | Reprompt | **0.066** | **0.094** | **0.132** | **0.454** |
|  | Goal | Baseline | 0.161 | 0.182 | 0.231 | 0.748 |
|  |  | Reprompt | **0.146** | **0.154** | **0.224** | **0.672** |

Table 6: Calibration error measurements using the 8-bit and 4-bit fine-tuned model version and 3 different LIBERO task suites, for 2 different methods of confidence estimation: (1) baseline (2) ensembling over semantically equivalent prompts ("Reprompt"). The prompt ensemble method improves all measures across all settings.

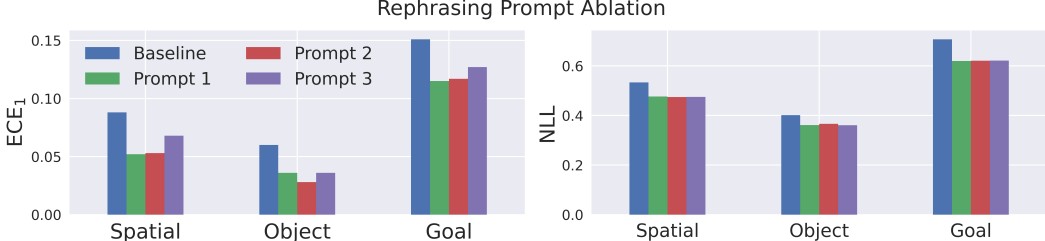

Figure 8: Ablation results for the prompt ensemble method, where different prompts are given to GPT-4o-mini for producing the rephrasings of the original instruction. Prompt 1 is used in the main experiments, while Prompts 2 and 3 are variants. Results show that improvements in calibration error from Reprompt are robust to different rephrasing prompts.

In addition to the prompt used to produce the results in Table 1, we try two additional prompts, recorded in Appendix Table 4; "Prompt 1" is the original prompt, and "Prompt 2" and "Prompt 3" represent new prompts for the ablation. Note that the different prompts lead to substantially different rephrasings, as Prompts 2 and 3 mandate the retention of the words "pick" and "place", while as Table 2 shows, Prompt 1 leads to these words being replaced. The ablation is run across all 3 task suites using fine-tuned OpenVLA, and we measure ECE$_1$ and NLL. Results are shown in Figure 8. Improvements in calibration error from the Reprompt method are robust to different rephrasing prompts, with all ensemble variants producing lower error than the baseline method across both metrics and all 3 task suites.

Second, we consider the effect of the number of prompts used in the prompt ensemble. Given a sound ensembling approach, we would expect to see improvement as more prompts are added to the ensemble (up to some point). To study this question, we record results using $k \in \{1, 5, 10, 20\}$ different rephrasings (randomly chosen for 1000 trials when $k < 20$). Results for the Spatial task suite are shown in Figure 9, and additional results for the Goal and Object task suites are in Appendix Figure 10. We see that the algorithm behaves favorably, where calibration error generally decreases as more instructions are included in the ensemble.

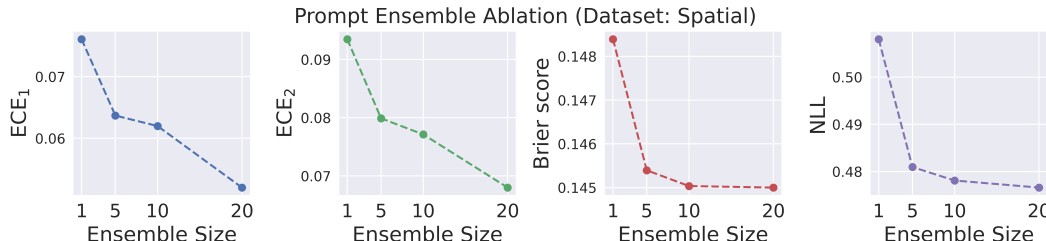

Figure 9: Ablating the number of prompts in the prompt ensemble for the Spatial task suite. Increasing the number of prompts generally improves calibration error.

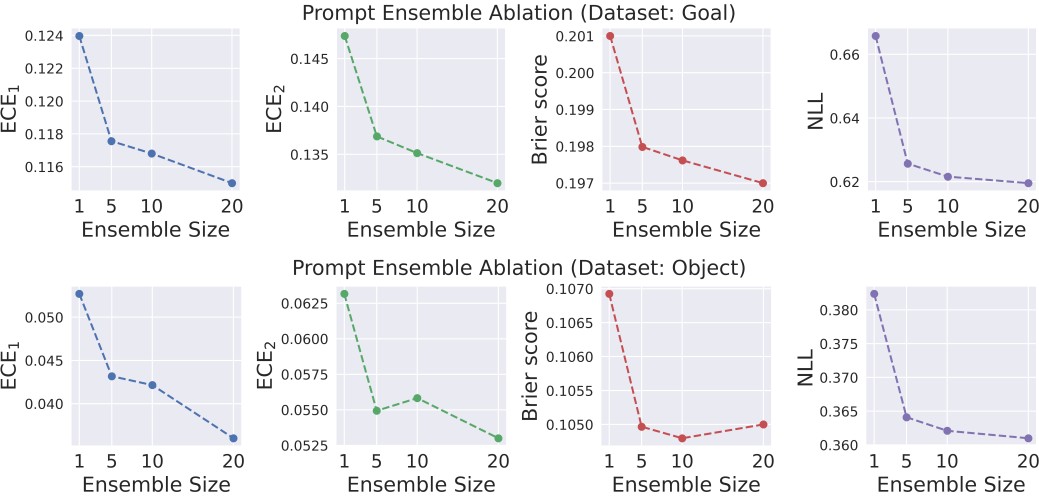

Figure 10: Ablating the number of prompts in the prompt ensemble for the Goal and Object task suites. Increasing the number of prompts generally improves calibration error.

### D.4    CALIBRATION ACROSS TASK TIME

Figures 11, 12, 13, 14, and 15 offer additional results on calibration across the task time horizon using the OpenVLA model. Figures 16, 17, and 18 offer results for the same experiment on UniVLA. Figure 19 shows further qualitative examples from the context-aware monitoring experiment.

#### D.4.1    QUALITATIVE EXAMPLES OF CONTEXT-AWARE CONFIDENCE MONITORING

To illustrate how context-aware confidence monitoring could work in practice, we apply it to a representative pick and place task from the Goal suite. The task is to "put the wine bottle on the rack", a case where the robot should be relatively conservative to avoid breaking glass. We consider a naive approach to context-aware monitoring, proposing to halt task performance when both:

1. The confidence level falls below a threshold set to the 10% quantile of confidence estimates for that point in the task horizon across all task trials (based on percent completion).

2. The robot is within a few inches of contacting an object, or already has contacted an object.

Since building a system to detect proximity to objects is beyond the scope of this work, we focus on a qualitative demonstration of this idea. For each example under examination, we plot current confidence for $\{0, 20, 40, 60, 80, 98\}\%$ completion, as well as the corresponding 10% quantile risk threshold and an image of the robot environment at that time.

Some particularly illustrative examples are shown in Figure 19. First, we consider the episode shown across the top row (Episode 29 of the 50 in the Goal task suite). Here, confidence begins high, but falls until it is slightly below the halting threshold at 20% task completion, coinciding with the

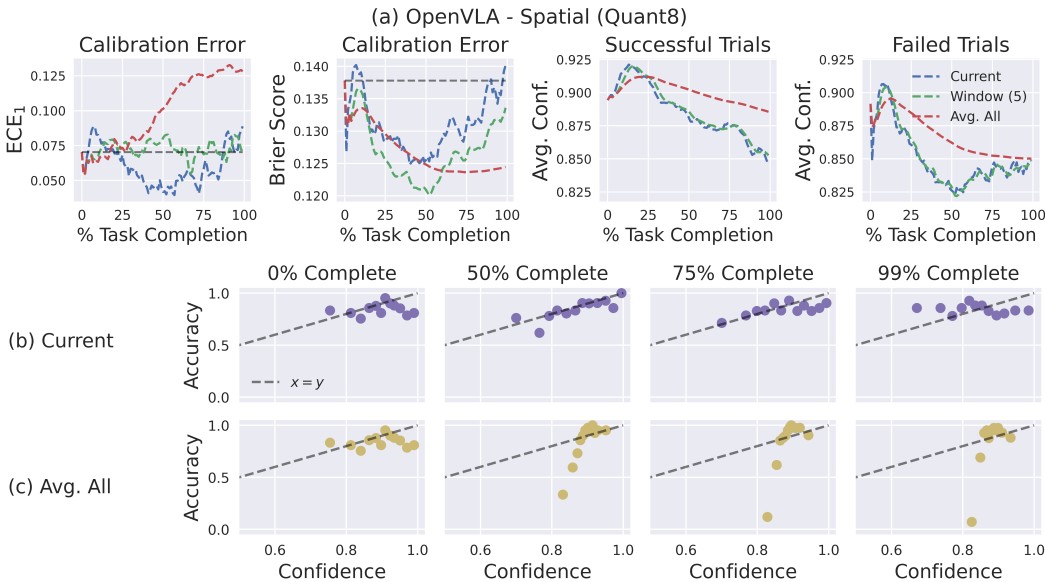

Figure 11: Empirical study of calibration error across task time horizon for the Spatial task suite and the Quant-8 OpenVLA model.

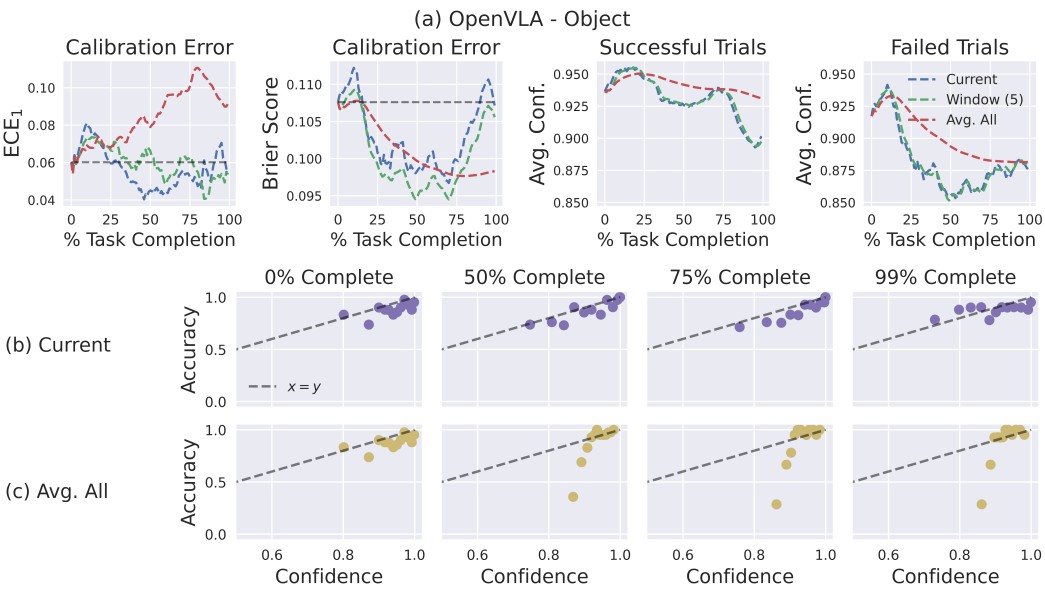

Figure 12: Empirical study of calibration error across task time horizon for the Object task suite and the full precision OpenVLA model.

difficult subtask of gripping the rounded glass bottle. Allowed to continue, the model is able to grip the bottle, and confidence rebounds as it approaches the wine rack. However, confidence once again falls quickly when it begins to set the bottle down, possibly because the difficult grip put the bottle in an unfavorable position for placement. In the end, the robot fails to place the bottle securely on the rack, with potentially negative real-world consequences. Given the opportunity, a safety intervention could have been performed at multiple points before this incident, either to avoid grasping the bottle in the first place or to reset the bottle safely on the table instead of trying a failed placement.

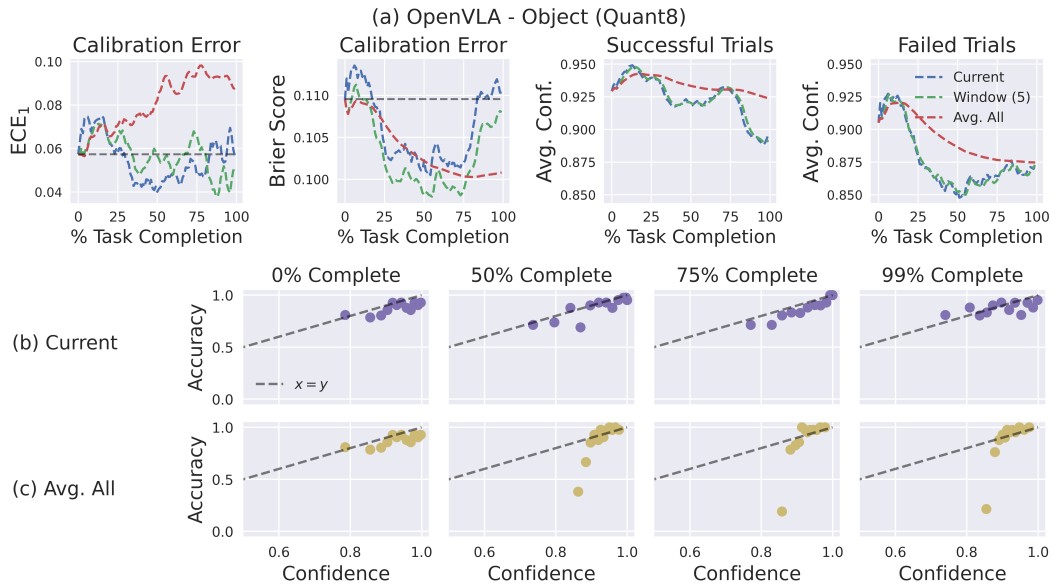

Figure 13: Empirical study of calibration error across task time horizon for the Object task suite and the Quant-8 OpenVLA model.

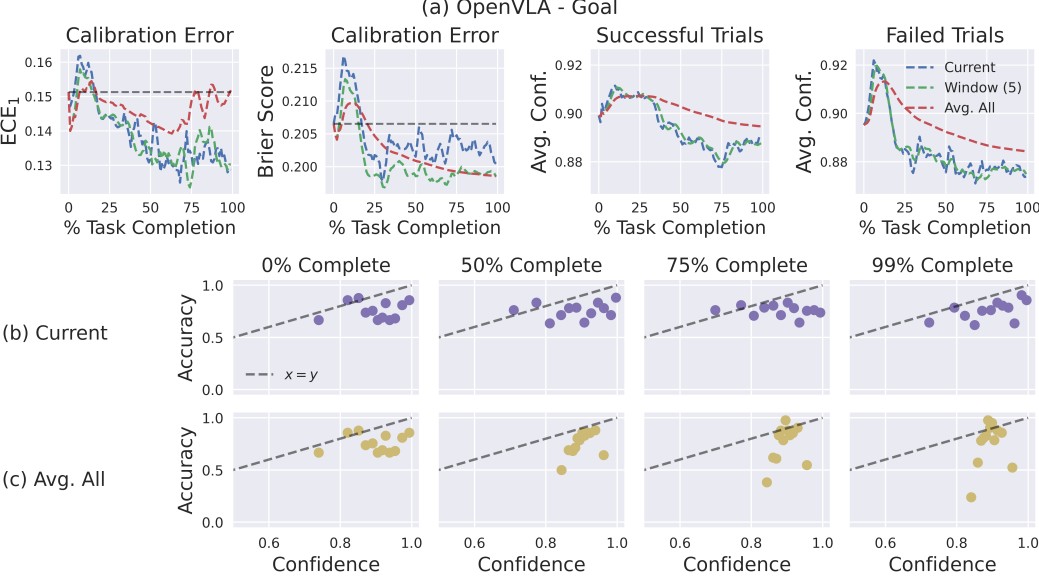

Figure 14: Empirical study of calibration error across task time horizon for the Goal task suite and the full precision OpenVLA model.

More such qualitative examples are provided in Figure 19. These include cases where the strategy succeeds and others where it fails (e.g., confidence falls below the threshold while grasping in an ultimately successful trial).

### D.5 CALIBRATION ACROSS ACTION DIMENSIONS

Figure 20 shows additional results for the action-wise recalibration experiments.

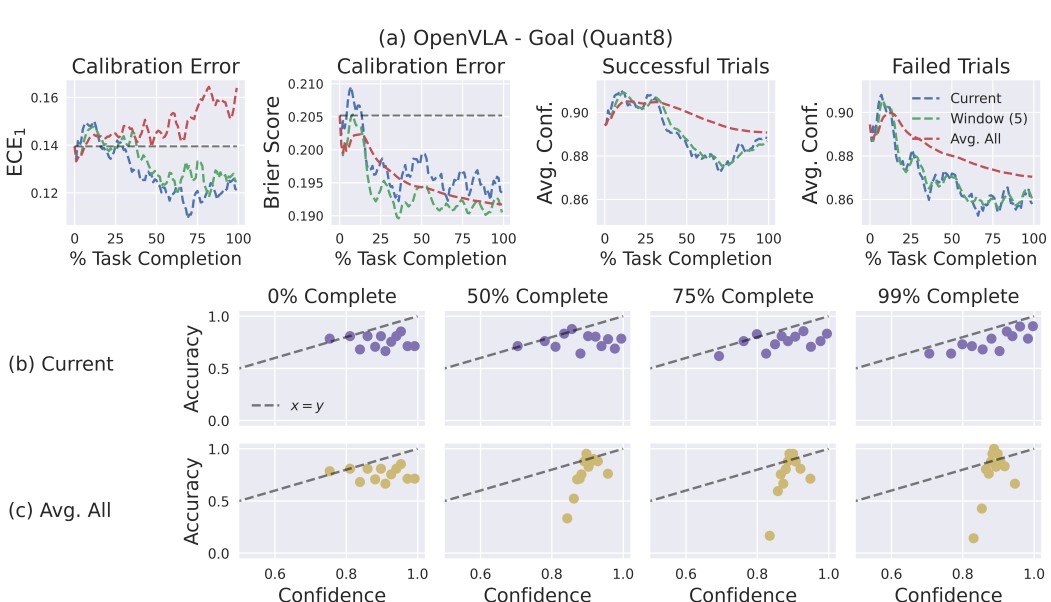

Figure 15: Empirical study of calibration error across task time horizon for the Goal task suite and the Quant-8 OpenVLA model.

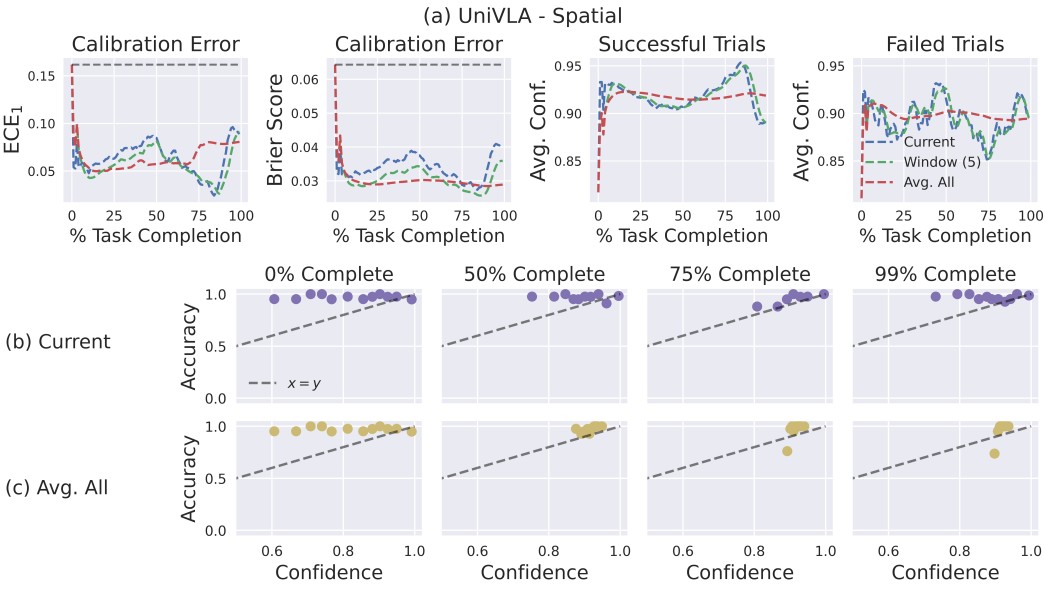

Figure 16: Empirical study of calibration error across task time horizon for the Spatial task suite and the UniVLA model.

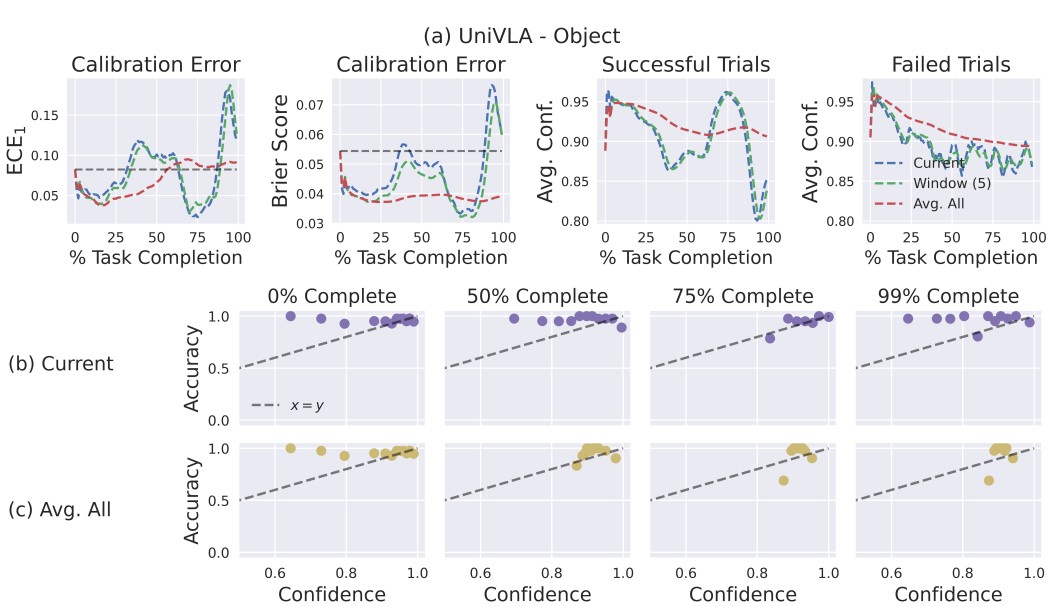

Figure 17: Empirical study of calibration error across task time horizon for the Object task suite and the UniVLA model.

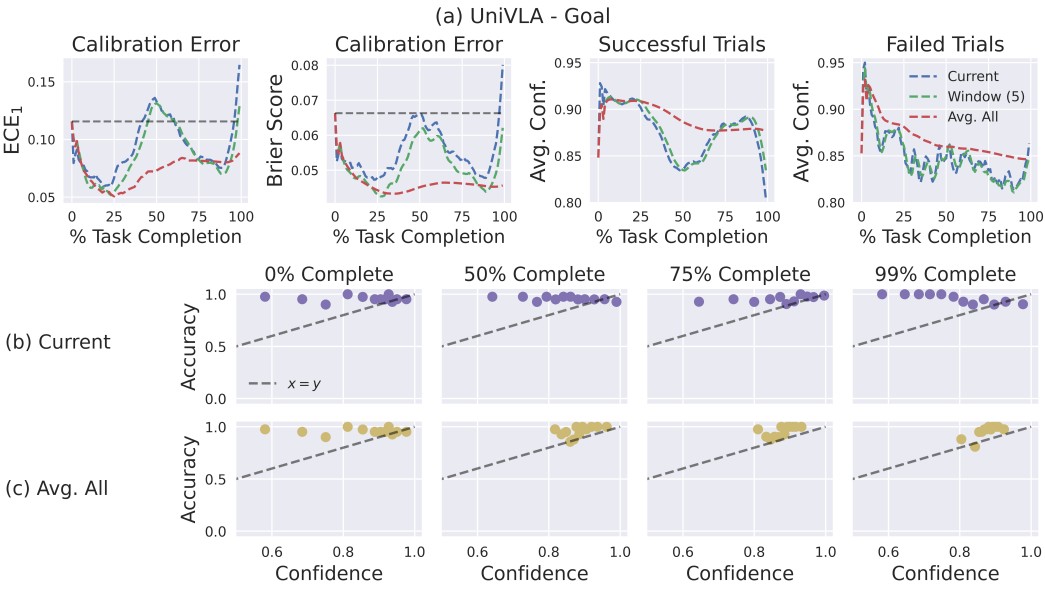

Figure 18: Empirical study of calibration error across task time horizon for the Goal task suite and the UniVLA model.

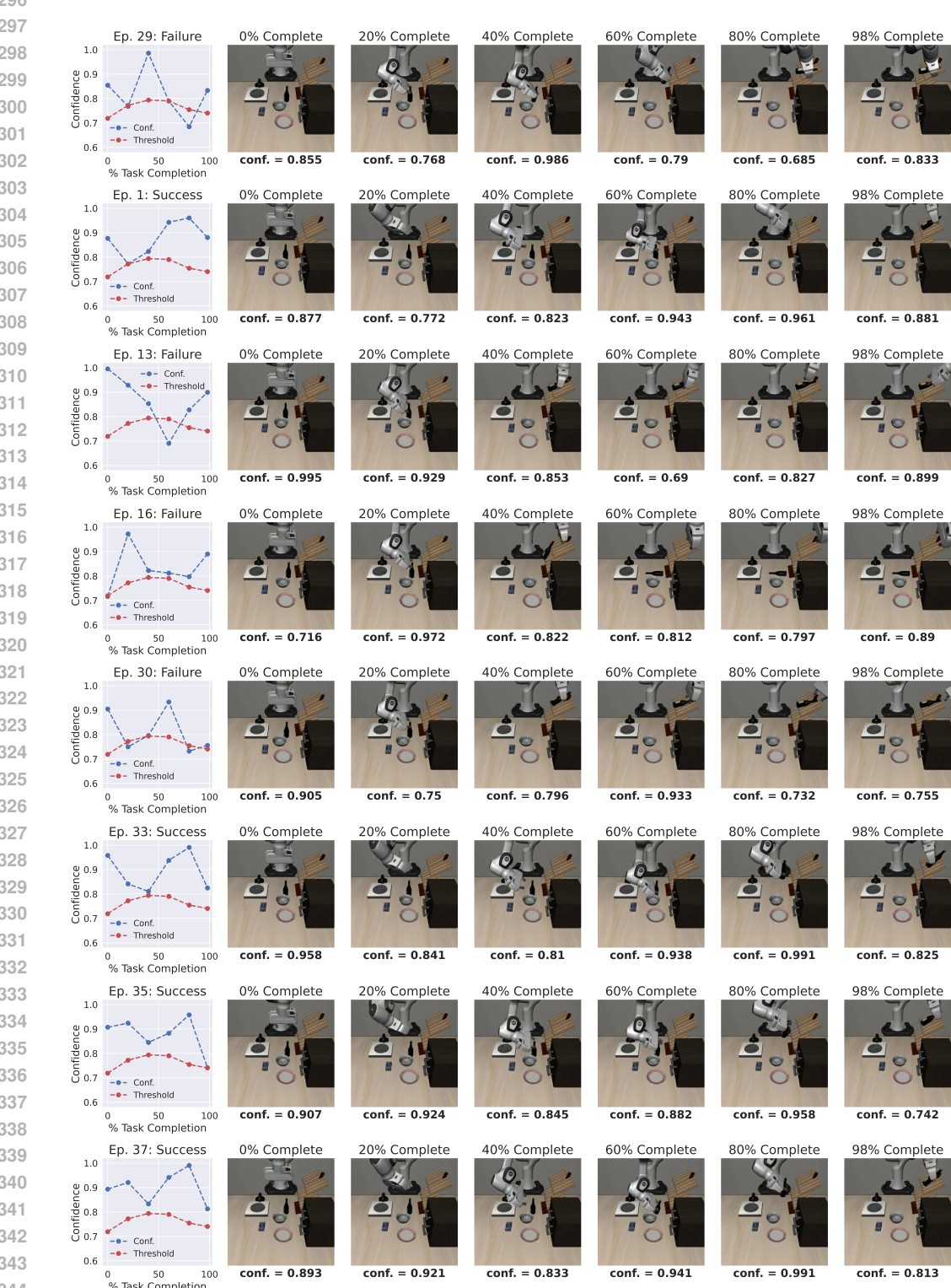

Figure 19: Qualitative examples of a context-aware confidence calibration strategy for the task "put the wine bottle on the rack". The red dashed line represents the 10% quantile of the confidence estimates output by the model across the task horizon, representing a potential threshold below which the robot may abstain from performing the task.

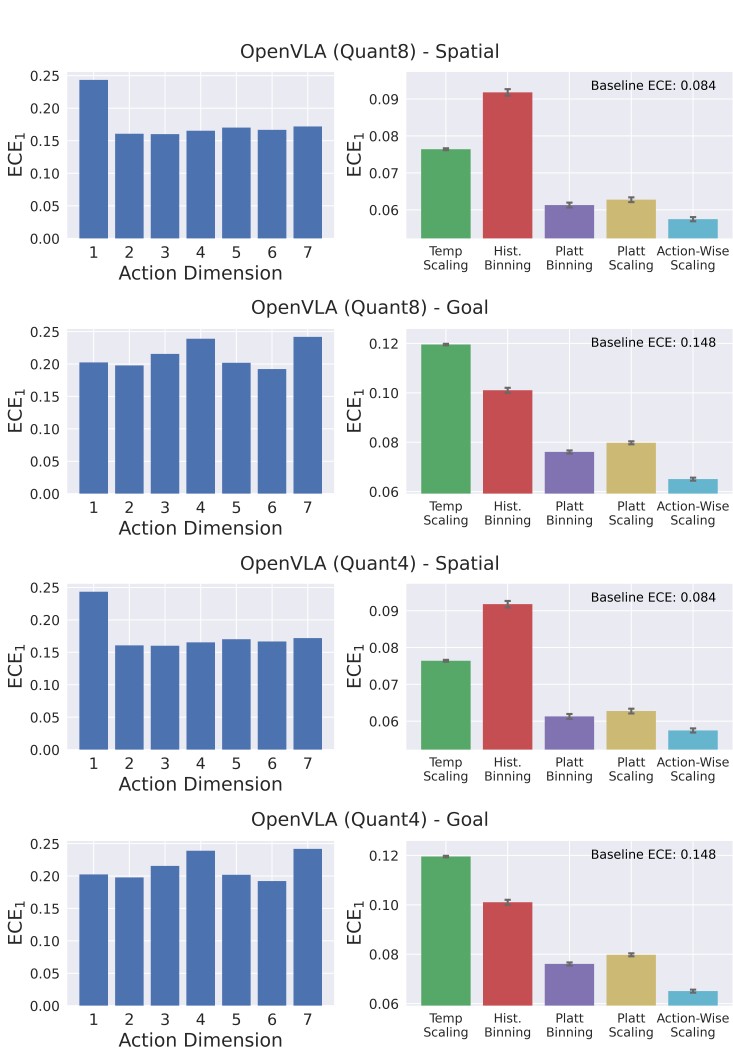

Figure 20: On the left of each pair of plots, we compare miscalibration across action dimensions. On the right, we compare the performance of typical Platt scaling, temperature scaling, histogram binning, and Platt binning to action-wise Platt scaling.

