# OpenReview forum: "Confidence Calibration in Vision-Language-Action Models"
_ICLR.cc/2026/Conference — Submitted to ICLR 2026_

### Official Review · Reviewer_tjhn · 2025-10-26

**Soundness:** 3
**Presentation:** 2
**Contribution:** 4
**Rating:** 8
**Confidence:** 4

**Summary:**

This paper is about calibration on vision-language-action models (VLAs). The authors explore the limitations of standard calibration/uncertainty techniques when applied to VLAs. Calibration has not really been explored for VLAs, making this the first work to touch the topic, and the paper makes contributions aiming to enable future research on VLA calibration.

The contributions are:
- An evaluation of success rate vs calibration error, finding that better task performance leads to better calibration error.
- Introduction of prompt ensembling for VLAs by making semantic modifications to the input prompt and taking average confidence.
- Extension of platt scaling to VLAs by applying platt scaling for each action dimension, and results showing that calibration does vary across action dimensions.
- Analysis of calibration over task time, like a time series, with interesting conclusions.

**Strengths:**

- The paper is well written and easy to follow.
- Calibration is an open problem in general for ML models and specifically for VLAs, I believe this combination has not been explored before and this paper makes novel contributions to enable future research on VLAs and uncertainty/calibration.
- The paper explores multiple aspects of calibration in VLAs, (1) defining calibration for VLAs as based on the confidence vs success rate, and then evaluating if success rate is related with calibration error. (2) Prompt ensembling for VLAs, by making semantic variations of the prompt, passing them through the model and then combining the predictions, this method is very well known in LLMs but this is the first time I see it applied to VLAs, and it does improve calibration error. (3) An extension of platt scaling for VLAs, noting that since a VLA outputs multiple action dimensions, the each dimension should be calibrated separately and this is supported by the experiments. (4) Analysis of calibration over task time, which treats the problem as a sequence of actions, each with a confidence score, and there are interesting insights here on how calibration error behaves across sequence points (what the authors call task completion).
- The experimental setup and experiments seem appropriate to me. The selection of baselines is also good (token probabilities) as well as using more modern uncertainty estimation methods like ensembles or platt scaling. The paper makes experiments on the LIBERO suite of tasks which is perfect for VLAs.
- Overall the contributions of this paper are strong and they open new venues for future research on VLA calibration.

**Weaknesses:**

- Some figures should be explained more clearly. Figure 4 in particular, the calibration plots in the two lower rows are not clear to me, what is the meaning of the dashed line? Since its a calibration plot I would expect the diagonal line to be perfect calibration but the dashed line does not match this expectation. Also the text in Sec 3.3, the text refers to figures using unclear terminology ("top row", etc), I think it is much better and readable to use figure numbers + letters to refer to subplots.
- Additionally about Figure 4, this result is about the spatial task suite, but this contains multiple instances of the spatial task, so its not clear what the plot represents, is it average task performance over task completion? Or median or another aggregation metric? I expect this to be clarified.
- Figure 6 makes comparisons between several methods but there is no comparison to the baseline method (token probabilities). This comparison should be made to put the ECE values into context.
- Overall the captions in this paper are not informative enough for the reader and should be improved, for example Figure 3 just mentions a comparison, but how should the reader interpret these plots? They compare different metrics between baseline and reprompt ensemble, which means values above the diagonal are improvements, but this is implicit and should be mentioned in the caption.

One minor weakness, I consider ECE to be mostly ECE_1, there is really no value in this paper on presenting ECE_2, it does not add any relevant information and it should be skipped.

**Questions:**

I only have one question, about the results about calibration on task time, it is not clear to me if results on Fig 4 generalize to all the task realizations inside the LIBERO spatial task suite? Could the authors clarify

---

> ### Author Response · Authors · 2025-11-20
> **Author Rebuttal**
>
> We thank the reviewer for their thorough and helpful feedback on our submission.  We are pleased that they see our work as a valuable contribution in the field of VLAs and VLA calibration.
>
> Below, please find responses to particular questions and comments about the work.  We have uploaded a revised draft of our submission with the requested changes, and we have also updated our experiments with results from 3 additional VLAs in order to sharpen our findings and understand their generality.
>
> Finally, please note that some Figures have changed numbering as we removed Figure 2 from the main paper given its redundancy with Table 1.  E.g., what was previously Figure 4 is now Figure 3, and what was previously 5 is now 4.
>
> **Some figures should be explained more clearly. Figure 4 in particular, the calibration plots in the two lower rows are not clear to me, what is the meaning of the dashed line? Since it’s a calibration plot I would expect the diagonal line to be perfect calibration but the dashed line does not match this expectation. Also the text in Sec 3.3, the text refers to figures using unclear terminology ("top row", etc), I think it is much better and readable to use figure numbers + letters to refer to subplots.**
>
> The dashed line represents perfect calibration, we think the reviewer may be confused by the scale and ratio of the plots.  We’ve changed the width/height ratio some, and added a legend with a label to clarify this.  We also agree that the references to the figures are not as clear as possible.  We have changed the plots and writing to refer to lettered sections of each of both Figure 4 and Figure 5 (now Figures 3+4).
>
> **Additionally about Figure 4, this result is about the spatial task suite, but this contains multiple instances of the spatial task, so its not clear what the plot represents, is it average task performance over task completion? Or median or another aggregation metric? I expect this to be clarified.**
>
> Results for each level of task completion are averaged across the 500 trials, in order to examine whether high-level confidence and calibration trends might occur.  We have added a sentence clarifying this.
>
> **Figure 6 makes comparisons between several methods but there is no comparison to the baseline method (token probabilities). This comparison should be made to put the ECE values into context.**
>
> Thank you for the suggestion, we’ve added the baseline ECE values to all recalibration plots.
>
> **Overall the captions in this paper are not informative enough for the reader and should be improved, for example Figure 3 just mentions a comparison, but how should the reader interpret these plots? They compare different metrics between baseline and reprompt ensemble, which means values above the diagonal are improvements, but this is implicit and should be mentioned in the caption.**
>
> We have clarified our captions to be more informative wherever possible.
>
> **One minor weakness, I consider ECE to be mostly ECE_1, there is really no value in this paper on presenting ECE_2, it does not add any relevant information and it should be skipped.**
>
> We agree that ECE is often measured via ECE-1. We would also highlight that there are some works, e.g., [1], [2], which focus mostly (or solely) on squared calibration error (ECE-2) over ECE-1, as well as many that measure both (e.g., [3]).  We included both for the sake of completeness.
>
>  - [1] https://arxiv.org/abs/1909.10155
>  - [2] https://arxiv.org/abs/2208.12084
>  - [3] https://arxiv.org/abs/2205.14334
>
> **I only have one question, about the results about calibration on task time, it is not clear to me if results on Fig 4 generalize to all the task realizations inside the LIBERO spatial task suite? Could the authors clarify**
>
> We do not expect these results to generalize to all the task realizations (by this we understand “single episode of a given task”) in a particular task suite.  These are high-level trends that emerge across many realizations and that might be used to guide the development of context- and risk-aware confidence monitoring strategies.  Illustrating this was an aim in examining individual cases via our qualitative experiment, as we highlight both examples where our proposed strategy works, and where it fails.  For example, in (the current) Figure 19, Episode 1 features a case where our rule would early-stop the robot when in fact it is a successful trial.  We have added a more explicit pointer to this in Line 422.  We hope that our study showing how a relatively simple rule can be developed and applied spurs further research on developing more complex, less heuristic strategies.
>
> Please let us know if this does not answer your question, as we would be happy to discuss further.

---

> ### Author Response · Authors · 2025-11-20
> **Rebuttal (ctd.)**
>
> ### New Experiment Results
>
> In order to strengthen our results, and broaden the conclusions that can be drawn from them, we have added results from 3 new VLA variants: MolmoAct, UniVLA, and NORA.  We have also added results for the LIBERO 10 task suite for all 4 VLAs under study, taking our total number of unique model/dataset combinations to 16 (plus the results from the quantized versions of OpenVLA, which make 22 total).  Here is an outline of the new experiment results and observations, which have also been integrated into the revised draft of our paper:
>
>  - Experiment 1 (Task success vs. calibration): We find that OpenVLA and MolmoAct show a roughly monotone relationship where higher success rates are associated with lower calibration error across all four metrics (ECE₁, ECE₂, Brier, NLL), whereas UniVLA and NORA display this trend only for Brier score and NLL but not for ECE. We point to particular differences in architecture and objectives as the potential source for these differences, and the need for future work with controlled ablations to fully understand which design choices drive the observed behavior.
>  - Experiment 2 (Prompt ensembles): We have added results from UniVLA and NORA to the prompt ensemble experiment, finding that this approach consistently improves calibration error.
>  - Experiment 3 (Calibration across task time): We have included results for UniVLA in this experiment, and mirrored our findings from OpenVLA where calibration error improves with task progress, pointing towards a significant opportunity to build context-aware confidence monitoring systems.
>  - Experiment 4 (Action-wise scaling): We have included results for MolmoAct (since UniVLA and NORA do not give dimension-wise confidence), as well as 2 more post hoc recalibration baselines (histogram binning and Platt binning).  Action-wise scaling consistently outperforms alternatives on both VLA variants.
>
> In summary, including the additional VLAs and task suite make the Experiment 1 findings more informative by exposing architecture-dependent calibration behavior, and they strengthen the original conclusions from Experiments 2-4 by demonstrating that our proposed remedies (prompt ensembles and action-wise scaling) apply across multiple VLA designs and tasks.
>
> We choose to focus on VLAs that produce discrete tokens with explicit logits, as this gives a natural notion of per-action probability and confidence. Models that predict continuous actions via flow matching (e.g., pi-0, pi0.5, FlowVLA, SmolVLA), diffusion (UnifiedVLA, GR00T N1), or L1 regression (VLA-Adapter, OpenVLA-OFT) would require additional methodological development to define and calibrate confidence, which we highlight as an interesting area for future work.

---

### Official Review · Reviewer_5yfq · 2025-10-26

**Soundness:** 2
**Presentation:** 2
**Contribution:** 2
**Rating:** 4
**Confidence:** 3

**Summary:**

The paper focuses on the problem of calibration in vision-language-action models (VLAs). VLAs are gaining popularity in the robotics research community due to multimodality. The paper seeks to measure the confidence of action predictions of VLA models.

The main contributions of this work are:

- evaluation to measure the relationship between task success and calibration error across multiple datasets and variants of the OpenVLA model.
- a lightweight, Bayesian-style method that averages a VLA’s confidence across semantically-similar rephrasings of prompts.
- an analysis of calibration over task time.
- The findings show that lower task performance is correlated with lower calibration scores.

**Strengths:**

- Originality: From my understanding of the literature, there are no previous works that studied the problem of calibration in VLA models. The authors focus on this problem, which has not been previously studied.
- Quality: Overall, the paper is decently presented with high-quality results for calibration scores in the OpenVLA setting. Looking at the code, it seems to be cleaned up well and is transparently released for reproducibility.
- Clarity: Overall, the paper is clear, but needs improvement around the claims around the comprehensiveness of the study (see the weaknesses here).
- Significance: I think this work has the potential to impact how VLA models are developed and improved in the community. The finding that task performance and calibration score are correlated is not all too surprising, but the impact of this finding is notable, especially as one might attempt a more targeted data collection.

**Weaknesses:**

- The authors claim to have a comprehensive evaluation, but this paper focuses exclusively on OpenVLA. Over the past year or so, various other VLA models have been made open-source, so evaluating on other open-source VLA models (Octo, Pi-0, etc) would significantly strengthen the claims in this work.
- As the authors acknowledge, all evaluation is performed in simulation, leaving out other sources of potential confidence miscalibration, including sensor noise, etc.

**Questions:**

- What is Figure 2 displaying? Which tasks are evaluated?
- The authors claim that the work is a “comprehensive evaluation to measure the
relationship between task success and calibration error across multiple datasets and VLA variants.” But what are all the VLA variants studied? Unless I misunderstood, this is referring to just the OpenVLA and its finetuned models, but I was expecting evaluation on other VLA models as well.
- Have the authors considered studying the benefits in calibration score measurement for targeted data collection and finetuning?

---

> ### Author Response · Authors · 2025-11-20
> **Author Rebuttal**
>
> We thank the reviewer for taking the time to review our submission, and offer valuable feedback.  Below we would like to respond to specific concerns raised by the review, in particular concerns around experimental scope.
>
> ### New Experiment Results
>
> In order to strengthen our results, and broaden the conclusions that can be drawn from them, we have added results from 3 new VLA variants: MolmoAct, UniVLA, and NORA.  We have also added results for the LIBERO 10 task suite for all 4 VLAs under study, taking our total number of unique model/dataset combinations to 16 (plus the results from the quantized versions of OpenVLA, which make 22 total).  Here is an outline of the new experiment results and observations, which have also been integrated into the revised draft of our paper:
>
>  - Experiment 1 (Task success vs. calibration): We find that OpenVLA and MolmoAct show a roughly monotone relationship where higher success rates are associated with lower calibration error across all four metrics (ECE₁, ECE₂, Brier, NLL), whereas UniVLA and NORA display this trend only for Brier score and NLL but not for ECE. We point to particular differences in architecture and objectives as the potential source for these differences, and the need for future work with controlled ablations to fully understand which design choices drive the observed behavior.
>  - Experiment 2 (Prompt ensembles): We have added results from UniVLA and NORA to the prompt ensemble experiment, finding that this approach consistently improves calibration error.
>  - Experiment 3 (Calibration across task time): We have included results for UniVLA in this experiment, and mirrored our findings from OpenVLA where calibration error improves with task progress, pointing towards a significant opportunity to build context-aware confidence monitoring systems.
>  - Experiment 4 (Action-wise scaling): We have included results for MolmoAct (since UniVLA and NORA do not give dimension-wise confidence), as well as 2 more post hoc recalibration baselines (histogram binning and Platt binning).  Action-wise scaling consistently outperforms alternatives on both VLA variants.
>
> In summary, including the additional VLAs and task suite make the Experiment 1 findings more informative by exposing architecture-dependent calibration behavior, and they strengthen the original conclusions from Experiments 2-4 by demonstrating that our proposed remedies (prompt ensembles and action-wise scaling) apply across multiple VLA designs and tasks.
>
> **The authors claim to have a comprehensive evaluation, but this paper focuses exclusively on OpenVLA. Over the past year or so, various other VLA models have been made open-source, so evaluating on other open-source VLA models (Octo, Pi-0, etc) would significantly strengthen the claims in this work.**
>
> We agree that evaluating additional VLAs strengthens the empirical scope of our study, and we have substantially expanded the experiments in the revised draft. In addition to OpenVLA, we now include three further VLAs with distinct architectures (MolmoAct, UniVLA, and NORA) and we add the LIBERO-10 suite for all four models. This yields 16 unique model/task-suite combinations (4 VLAs x 4 suites) plus 6 additional settings for 8-bit and 4-bit OpenVLA, for a total of 22 model/dataset combinations.
>
> We choose to focus on VLAs that produce discrete tokens with explicit logits, as this gives a natural notion of per-action probability and confidence. Models that predict continuous actions via flow matching (e.g., pi-0, pi0.5, FlowVLA, SmolVLA), diffusion (UnifiedVLA, GR00T N1), or L1 regression (VLA-Adapter, OpenVLA-OFT) would require additional methodological development to define and calibrate confidence, which we highlight as an interesting area for future work.

---

> > ### Author Response · Authors · 2025-11-20
> > **Rebuttal (ctd.)**
> >
> > **As the authors acknowledge, all evaluation is performed in simulation, leaving out other sources of potential confidence miscalibration, including sensor noise, etc.**
> >
> > Studying calibration across a broader set of environments, including real robots, is an important direction. In this work we deliberately standardize on LIBERO because it is, to our knowledge, the dominant open-source benchmark for modern VLAs and the only one for which a wide range of fine-tuned models are publicly released. LIBERO is used to evaluate UnifiedVLA, FlowVLA, SmolVLA, OpenVLA, OpenVLA-OFT, VLA-Adapter, UniVLA, MolmoAct, Nora, pi-0, pi0-FAST, and pi-0.5, among others, whereas no other simulator appears in more than a small handful of these papers. Focusing on LIBERO lets us compare calibration behavior across multiple independently developed VLAs in a common, widely adopted setting.
> >
> > Regarding real-world experiments, our choice of simulation is driven by the sample complexity of calibration.  Our results feature measurements from more than 100,000 robot episodes.  On the other hand, prominent works from well-resourced research labs like Physical Intelligence (pi-0, pi-0.5) and Nvidia (GR00T N1) evaluate on the order of 100 real-world episodes.  This amount of data would not be enough to produce a single reliable ECE measurement.  We have added a further note about this to our limitations, pointing out that studying these techniques on real robots will ultimately be essential for understanding how various factors affect VLA calibration under realistic conditions.  Finally, we highlight that this choice of simulation-only evaluation is not uncommon in robotics research.
> >
> > **What is Figure 2 displaying? Which tasks are evaluated?**
> >
> > Figure 2 features a visualization of task error rates compared against 4 different calibration error measurements for 4 VLA variants (OpenVLA, MolmoAct, UniVLA, and NORA) and 4 LIBERO task suites (Spatial, Object, Goal, 10), as well as OpenVLA 8- and 4-bit versions on Spatial, Object, and Goal, for a total of 22 measurements per plot.  We have clarified this in our revised submission.
> >
> > **The authors claim that the work is a “comprehensive evaluation to measure the relationship between task success and calibration error across multiple datasets and VLA variants.” But what are all the VLA variants studied? Unless I misunderstood, this is referring to just the OpenVLA and its finetuned models, but I was expecting evaluation on other VLA models as well.**
> >
> > We have replaced the word “comprehensive” with “first-of-its-kind”, and included results from 3 additional VLA models.  Please see above for results on these models, as well as an explanation of their choosing.
> >
> > **Have the authors considered studying the benefits in calibration score measurement for targeted data collection and finetuning?**
> >
> > We agree that a natural application or extension of this work is using confidence estimates to guide efficient collection of data/demonstrations, as in active learning.  We have added this point to our Future Work section in Appendix B, and referenced it specifically in the final sentence of our introduction.

---

> > > ### Author Response · Authors · 2025-11-26
> > > **rebuttal follow up**
> > >
> > > Hello, we just wanted to politely follow up here, and see if the reviewer has any remaining questions that we might be able to address.  We hope that our addition of results from 3 new VLAs (as well as 1 additional task suite) was able to address the reviewer’s main concerns.
> > >
> > > Thank you again for taking the time to review our submission.

---

### Official Review · Reviewer_EuHf · 2025-10-27

**Soundness:** 3
**Presentation:** 4
**Contribution:** 3
**Rating:** 6
**Confidence:** 2

**Summary:**

This paper studies confidence calibration in Vision-Language-Action (VLA) models, a new class of large multimodal foundation models that connect visual perception, natural language, and control. The authors identify that while these models are becoming central to embodied AI, their confidence outputs are not well understood. They present the first systematic calibration analysis for VLAs using OpenVLA on the LIBERO benchmark and explore how calibration relates to task success, how it evolves during execution, and how to improve it through simple post-hoc methods. They propose three lightweight techniques and show that these substantially improve calibration without retraining. The study is novel and relevant; the methodology is clear, and the metrics are well-documented.

**Strengths:**

The paper is clear, well-structured, and grounded in the calibration literature. The experimental design is sound, using multiple metrics (ECE, Brier, NLL) and analyzing calibration across tasks, time, and model precision. The results are consistent and interpretable. For example, better-performing models are also better calibrated, and early overconfidence decreases over time. The proposed fixes are simple but effective and practical for real-world systems.

**Weaknesses:**

- Methods are empirical adaptations rather than theoretical advances.
- All experiments are simulation-only; no real-robot validation. More generally, it is not clear what role robotics plays in this framework.
- Only one model family (OpenVLA) and one benchmark were tested.
- Missing comparisons to other post-hoc calibration baselines.
- Discussion of broader implications (safety, planning) could be deeper.
- The evaluation is entirely simulation-based and limited to OpenVLA on LIBERO tasks, which restricts how confidently the conclusions can be generalized. The authors should demonstrate calibration performance on at least one real-robot setup or different VLA architecture (e.g., RT-2, π₀) to validate the robustness of their approach.
-  The reliance on one benchmark makes it unclear whether the observed improvements hold under sensor noise, dynamic lighting, or embodiment variation, all critical for real deployment.
- The study also lacks ablation or sensitivity analysis on the number of prompt rephrasings, calibration-set size, and hyperparameters of the scaling functions, which could expose overfitting or instability.
- Finally, stronger comparisons against standard post-hoc baselines would make the results more convincing. Without these additions, the empirical evaluation, while neat and well-presented, feels too narrow and over-controlled for a paper claiming to improve confidence calibration for “real-world embodied systems.”


Minor comments:
--Include error bars for ECE and Brier results.
--Clarify binning strategy for ECE.
--Discuss briefly how calibration could be integrated during training.
--Expand limitations to connect better with real-world deployment.

**Questions:**

- What is the role the fact that this is a robotics system plays in the suggested approach? It seems to be a general approach that was tested on robotics.

- In the examined setting, does it make sense to only consider the most likely action and not consider a more robust version?

- What do you expect to be the core sim-to-real hurdles this framework would address (beyond the cost of evaluating real-world robotic systems)?

---

> ### Author Response · Authors · 2025-11-20
> **Author Rebuttal**
>
> We thank the reviewer for the time taken in reviewing and offering feedback on our manuscript.  Below we would like to respond to particular concerns raised by the review.
>
> ### New Experiment Results
>
> In order to strengthen our results, and broaden the conclusions that can be drawn from them, we have added results from 3 new VLA variants: MolmoAct, UniVLA, and NORA.  We have also added results for the LIBERO 10 task suite for all 4 VLAs under study, taking our total number of unique model/dataset combinations to 16 (plus the results from the quantized versions of OpenVLA, which make 22 total).  Here is an outline of the new experiment results and observations, which have also been integrated into the revised draft of our paper:
>
>  - Experiment 1 (Task success vs. calibration): We find that OpenVLA and MolmoAct show a roughly monotone relationship where higher success rates are associated with lower calibration error across all four metrics (ECE₁, ECE₂, Brier, NLL), whereas UniVLA and NORA display this trend only for Brier score and NLL but not for ECE. We point to particular differences in architecture and objectives as the potential source for these differences, and the need for future work with controlled ablations to fully understand which design choices drive the observed behavior.
>  - Experiment 2 (Prompt ensembles): We have added results from UniVLA and NORA to the prompt ensemble experiment, finding that this approach consistently improves calibration error.
>  - Experiment 3 (Calibration across task time): We have included results for UniVLA in this experiment, and mirrored our findings from OpenVLA where calibration error improves with task progress, pointing towards a significant opportunity to build context-aware confidence monitoring systems.
>  - Experiment 4 (Action-wise scaling): We have included results for MolmoAct (since UniVLA and NORA do not give dimension-wise confidence), as well as 2 more post hoc recalibration baselines (histogram binning and Platt binning).  Action-wise scaling consistently outperforms alternatives on both VLA variants.
>
> In summary, including the additional VLAs and task suite make the Experiment 1 findings more informative by exposing architecture-dependent calibration behavior, and they strengthen the original conclusions from Experiments 2-4 by demonstrating that our proposed remedies (prompt ensembles and action-wise scaling) apply across multiple VLA designs and tasks.

---

> > ### Author Response · Authors · 2025-11-20
> > **Rebuttal (ctd.)**
> >
> > **Methods are empirical adaptations rather than theoretical advances.**
> >
> > We agree that our proposed methods build on established ideas (ensembling and post-hoc recalibration), and we make no claims to theoretical advances. That said, we believe the paper makes multiple domain-specific technical contributions:
> >  - **First calibration study for VLAs.** We introduce, to our knowledge, the first formulation for benchmarking confidence calibration in vision-language-action models, including a concrete procedure for extracting effective scalar confidences, and a framework that analyzes calibration as a function of task success, episode progress, and action dimensions. This goes beyond prior work on classification/LLMs, where the prediction structure and evaluation setting are quite different.
> >  - **Prompt ensembles for VLAs.** Our “prompt ensemble” method exploits a specifically VLA-relevant latent variable, the phrasing of the natural-language instruction, to implement a Bayesian-motivated averaging over semantically equivalent prompts. We show that this design consistently reduces ECE by >20% on average across models and task suites, while leaving the policy unchanged and being practical to deploy with batched inference.
> >  - **Action-wise Platt scaling.** Existing post-hoc recalibration approaches are generally meant for classification models and treat each prediction as a single scalar probability. VLAs, by contrast, emit one probability distribution per action dimension. We propose action-wise Platt scaling, which learns separate recalibration maps per action dimension and then aggregates them. Empirically, this consistently outperforms global Platt scaling, temperature scaling, and histogram-based methods, and reveals that different degrees of freedom can be miscalibrated in different ways.
> >
> > More broadly, our work is in the same spirit as (possibly the most) influential deep learning UQ papers such as Guo et al. (2017) [1], Lakshminarayanan et al. (2017) [2], and Gal & Ghahramani (2016) [3]: the novelty lies not in inventing entirely new mathematical machinery and more in (i) identifying a previously unstudied but important setting (VLA calibration), (ii) adapting and extending simple tools in a way that respects its structure (instruction-level ensembles and action-wise recalibration), and (iii) uncovering new empirical phenomena (e.g., improved calibration mid-trajectory and heterogeneous calibration across action dimensions) that we hope will guide future, more sophisticated methods.
> >
> >  - [1] On Calibration of Modern Neural Networks (https://arxiv.org/abs/1706.04599) - Introduces temperature scaling, a simple extension of Platt Scaling to a multi-class setting.
> >  - [2] Simple and Scalable Predictive Uncertainty Estimation using Deep Ensembles (https://arxiv.org/abs/1612.01474) - show that a straightforward combination of standard tools (proper scoring rules, independent ensembles, and adversarial training) yields strong predictive uncertainty.
> >  - [3] Dropout as a Bayesian Approximation: Representing Model Uncertainty in Deep Learning (https://arxiv.org/abs/1506.02142) - Show that dropout at test time yields well-behaved predictive uncertainties with minimal changes to training or architecture.
> >
> > **Focus on LIBERO simulation environment**
> >
> > Studying calibration across a broader set of environments, including real robots, is an important direction. In this work we deliberately standardize on LIBERO because it is, to our knowledge, the dominant open-source benchmark for modern VLAs and the only one for which a wide range of fine-tuned models are publicly released. LIBERO is used to evaluate UnifiedVLA, FlowVLA, SmolVLA, OpenVLA, OpenVLA-OFT, VLA-Adapter, UniVLA, MolmoAct, Nora, pi-0, pi0-FAST, and pi-0.5, among others, whereas no other simulator appears in more than a small handful of these papers. Focusing on LIBERO lets us compare calibration behavior across multiple independently developed VLAs in a common, widely adopted setting.
> >
> > Regarding real-world experiments, our choice of simulation is driven by the sample complexity of calibration.  Our results feature measurements from more than 100,000 robot episodes.  On the other hand, prominent works from well-resourced research labs like Physical Intelligence (pi-0, pi-0.5) and Nvidia (GR00T N1) evaluate on the order of 100 real-world episodes.  This amount of data would not be enough to produce a single reliable ECE measurement.  We have added a further note about this to our limitations, pointing out that studying these techniques on real robots will ultimately be essential for understanding how various factors affect VLA calibration under realistic conditions.  Finally, we highlight that this choice of simulation-only evaluation is not uncommon in robotics research.

---

> > > ### Author Response · Authors · 2025-11-20
> > > **Rebuttal (ctd.)**
> > >
> > > **Inclusion of other model families beyond OpenVLA**
> > >
> > > We agree that evaluating additional VLAs strengthens the empirical scope of our study, and we have substantially expanded the experiments in the revised draft. In addition to OpenVLA, we now include three further VLAs with distinct architectures (MolmoAct, UniVLA, and NORA) and we add the LIBERO-10 suite for all four models. This yields 16 unique model/task-suite combinations (4 VLAs x 4 suites) plus 6 additional settings for 8-bit and 4-bit OpenVLA, for a total of 22 model/dataset combinations.
> > >
> > > We choose to focus on VLAs that produce discrete tokens with explicit logits, as this gives a natural notion of per-action probability and confidence. Models that predict continuous actions via flow matching (e.g., pi-0, pi0.5, FlowVLA, SmolVLA), diffusion (UnifiedVLA, GR00T N1), or L1 regression (VLA-Adapter, OpenVLA-OFT) would require additional methodological development to define and calibrate confidence, which we highlight as an interesting area for future work.
> > >
> > > **Discussion of broader implications (safety, planning) could be deeper.**
> > >
> > > As this is the first work considering the calibration of VLAs, and we are constrained by space, we chose to focus on high-level concerns, such as methods for producing and measuring confidence estimates, and presentation of diverse experimental results.  While we highlight these concerns around safety in the introduction, methods, and experiments sections, we also point out in the Future Work section that there remain many open challenges in designing adaptive risk-aware robotic pipelines, e.g., via selective execution, planning, or human-in-the-loop systems.
> > >
> > > **The study also lacks ablation or sensitivity analysis on the number of prompt rephrasings, calibration-set size, and hyperparameters of the scaling functions, which could expose overfitting or instability.**
> > >
> > > We do perform ablations of the prompt ensemble approach along multiple dimensions, including number of rephrasing.  Please see Line 299 in our original manuscript.  First, we consider the effect of changing the prompt given to GPT-4o-mini for producing the 20 instruction rephrasings, finding that improvements in calibration error from the Reprompt method are robust to different rephrasing prompts. Second, we consider the effect of the number of prompts used in the prompt ensemble, and see that the algorithm behaves favorably, where calibration error generally decreases as more instructions are included in the ensemble.
> > >
> > > With regard to recalibration, standard temperature scaling and platt scaling introduces no additional hyperparameters, nor does our proposed action-wise scaling variant.
> > >
> > > **Stronger comparisons against standard post-hoc baselines would make the results more convincing. Without these additions, the empirical evaluation, while neat and well-presented, feels too narrow and over-controlled for a paper claiming to improve confidence calibration for “real-world embodied systems.”**
> > >
> > > Thank you for the suggestion.  We have included more post-hoc calibration baselines, in particular Histogram Binning and Platt Binning.  We are unable to locate the quote about “real-world embodied systems” in our submission, but if the reviewer has specific claims from the paper around recalibration that they would like us to revise or revisit, we would be happy to do so.  Also, we have added an additional line to our limitations section highlighting the need to perform these experiments on real robots in the real world.
> > >
> > > **Clarify binning strategy for ECE.**
> > >
> > > To calculate ECE, we use 12 equal-mass bins and the Python package released with Kumar et al. (2019).  This is reported in line 215 of our submission.
> > >
> > > **Discuss briefly how calibration could be integrated during training.**
> > >
> > > In a previous era of calibration research computer vision researchers proposed training interventions to augment the cross-entropy loss with other objectives seeking improved calibration, potentially at a cost to overall accuracy.  However, subsequent research found that, in fact, more accurate networks are generally better calibrated (and easier to recalibrate), and thus modifications to training procedures may not be needed (Minderer et al., 2021). Instead, techniques such as post hoc recalibration (Guo et al., 2017; Zadrozny & Elkan, 2001; Kumar et al., 2019) and ensembling (Lakshminarayanan et al., 2017; Fort & Lakshminarayanan, 2024), applied to models trained for high accuracy, are sufficient to achieve low calibration error (although calibration under distribution shift remains a significant challenge (Ovadia et al., 2019)).
> > >
> > > **Expand limitations to connect better with real-world deployment.**
> > >
> > > We have added further discussion in the limitations section of the need to perform further experiments under real-world conditions.

---

> > > > ### Author Response · Authors · 2025-11-20
> > > >
> > > > **What is the role the fact that this is a robotics system plays in the suggested approach? It seems to be a general approach that was tested on robotics.**
> > > >
> > > > Our formalization is given with respect to a VLA model, all experiments are performed using a popular robotics simulation platform, and examples of where robotics is considered in specific methodological and experimental choices include:
> > > >  - The first formulation for benchmarking confidence calibration in vision-language-action models, including the first concrete procedure for extracting effective scalar confidences.
> > > >  - Our prompt ensemble method exploits a specifically VLA-relevant latent variable, the phrasing of the natural-language instruction, to implement a Bayesian-motivated averaging over semantically equivalent prompts.
> > > >  - We propose action-wise Platt scaling, which learns separate recalibration maps per action dimension and then aggregates them, while existing post-hoc recalibration approaches are generally meant for classification models and treat each prediction as a single scalar probability.
> > > >
> > > > If the reviewer has questions about particular aspects of the formalization, methods, or experiments, we would be happy to answer further, but we feel that the focus on robotics is apparent, and none of these methods or choices are clearly useful in or directly taken from any other domain.
> > > >
> > > > **In the examined setting, does it make sense to only consider the most likely action and not consider a more robust version?**
> > > >
> > > > We made this baseline choice in keeping with prominent findings from other related domains.  For example, the standard method for confidence estimation in computer vision derives its estimate from the top-class probability.  Similarly, LLM UQ approaches often rely on top-token probabilities.  We believe that our experiments validate that this is a reasonable choice, as we produce many non-trivial calibration results.  We leave it to future work to consider other confidence estimation baselines.
> > > >
> > > > **What do you expect to be the core sim-to-real hurdles this framework would address (beyond the cost of evaluating real-world robotic systems)?**
> > > >
> > > > The goal of this framework is not to directly address sim-to-real hurdles.  However, uncertainty quantification is a core tool for effecting trustworthy machine learning deployments, and such tools are especially critical in real-world deployments, and in cases where reliability is inherently difficult (e.g., when the model is trained on data from a different distribution).

---

> > > > > ### Author Response · Authors · 2025-11-26
> > > > >
> > > > > Hello, we just wanted to politely follow up here, and see if the reviewer has any remaining questions that we might be able to address.  We hope that our addition of results from 3 new VLAs and the other changes we have made to our submission were able to address the reviewer’s main concerns.
> > > > >
> > > > > Thank you again for taking the time to review our submission.

---

### Official Review · Reviewer_Cqir · 2025-10-31

**Soundness:** 1
**Presentation:** 2
**Contribution:** 1
**Rating:** 2
**Confidence:** 4

**Summary:**

The paper presents a study of confidence calibration of vision-language action models. It analyzes the correlation between model confidence and task success rate, as well as model confidence across different perturbed language instructions. Experiments on LIBERO with OpenVLA demonstrate that calibration error decreases with higher task success rates and longer trajectories.

**Strengths:**

- The paper is well-motivated. The problem of confidence calibration is highly relevant for deploying trustworthy robotic systems in high-stakes, real-world environments

**Weaknesses:**

- The paper's technical novelty is limited. The proposed methods are largely simple applications of existing techniques to an existing VLA.
### The claim of a systematic study is not well-supported by the experiments:
- All experiments are conducted on a single VLA, OpenVLA. This is not enough to claim a systematic study of calibration errors for VLAs. Diffusion-based VLAs might exhibit different calibration behavior.
- All experiments are conducted in the LIBERO simulation. For a study on this topic, the lack of any diverse simulation (or real-world) experiments is a significant weakness.
- The paper's default choice of using the pre-action confidence (ci,1​) as the primary metric is not well-justified. OpenVLA is only trained on single-image observations and produces a single action. It has no explicit understanding of task progress. Thus, it is unclear if the confidence at the very first step is representative of the entire trajectory's confidence. Further ablations are required to validate this choice.
- No experiments on zero-shot or out-of-domain environments, and the relation to model confidence
- Other aggregation methods are not evaluated or ablated

In its current state, I think the paper is not ready for publication at ICLR. While the problem of VLA calibration is highly relevant to robotics and underexplored, the paper's technical novelty is somewhat limited. More importantly, the experimental scope is not sufficient to support the claims of a systematic study.

**Questions:**

- What is the reasoning behind choosing the first frame's confidence as the default metric? Did you experiment with other aggregation metrics?
- What does the calibration look like in OOD scenarios? e.g., with new objects or unseen settings?
- What does the confidence look like without model fine-tuning?
- Why do you compare against quantized versions of the same model?

---

> ### Author Response · Authors · 2025-11-20
> **Author Rebuttal**
>
> We thank the reviewer for the time taken in reviewing and offering feedback on our submission.  Below please find our responses to your individual concerns, including a significant addition of results for 3 more VLA variants and 1 more task suite.  Also, we have integrated these changes into a revised submission.  Please note that some line numbers and figure numbers have changed between our original submission and this revised draft.
>
> ### New Experiment Results
>
> In order to strengthen our results, and broaden the conclusions that can be drawn from them, we have added results from 3 new VLA variants: MolmoAct, UniVLA, and NORA.  We have also added results for the LIBERO 10 task suite for all 4 VLAs under study, taking our total number of unique model/dataset combinations to 16 (plus the results from the quantized versions of OpenVLA, which make 22 total).  Here is an outline of the new experiment results and observations, which have also been integrated into the revised draft of our paper:
>
>  - Experiment 1 (Task success vs. calibration): We find that OpenVLA and MolmoAct show a roughly monotone relationship where higher success rates are associated with lower calibration error across all four metrics (ECE₁, ECE₂, Brier, NLL), whereas UniVLA and NORA display this trend only for Brier score and NLL but not for ECE. We point to particular differences in architecture and objectives as the potential source for these differences, and the need for future work with controlled ablations to fully understand which design choices drive the observed behavior.
>  - Experiment 2 (Prompt ensembles): We have added results from UniVLA and NORA to the prompt ensemble experiment, finding that this approach consistently improves calibration error.
>  - Experiment 3 (Calibration across task time): We have included results for UniVLA in this experiment, and mirrored our findings from OpenVLA where calibration error improves with task progress, pointing towards a significant opportunity to build context-aware confidence monitoring systems.
>  - Experiment 4 (Action-wise scaling): We have included results for MolmoAct (since UniVLA and NORA do not give dimension-wise confidence), as well as 2 more post hoc recalibration baselines (histogram binning and Platt binning).  Action-wise scaling consistently outperforms alternatives on both VLA variants.
>
> In summary, including the additional VLAs and task suite make the Experiment 1 findings more informative by exposing architecture-dependent calibration behavior, and they strengthen the original conclusions from Experiments 2-4 by demonstrating that our proposed remedies (prompt ensembles and action-wise scaling) apply across multiple VLA designs and tasks.

---

> > ### Author Response · Authors · 2025-11-20
> > **Rebuttal (ctd.)**
> >
> > **The paper's technical novelty is limited. The proposed methods are largely simple applications of existing techniques to an existing VLA.**
> >
> > While our proposed methods build on established ideas (ensembling and post-hoc recalibration), we believe the paper makes multiple domain-specific technical contributions beyond a straightforward application of existing techniques.
> >
> > - First calibration study for VLAs: We introduce, to our knowledge, the first formulation for benchmarking confidence calibration in vision-language-action models, including a concrete procedure for extracting effective scalar confidences, and a framework that analyzes calibration as a function of task success, episode progress, and action dimensions. This goes beyond prior work on classification/LLMs, where the prediction structure and evaluation setting are quite different.
> > - Prompt ensembles for VLAs: Our “prompt ensemble” method exploits a specifically VLA-relevant latent variable, the phrasing of the natural-language instruction, to implement a Bayesian-motivated averaging over semantically equivalent prompts. We show that this design consistently reduces ECE by >20% on average across models and task suites, while leaving the policy unchanged and being practical to deploy with batched inference.
> > - Action-wise Platt scaling: Existing post-hoc recalibration approaches are generally meant for classification models and treat each prediction as a single scalar probability. VLAs, by contrast, emit one probability distribution per action dimension. We propose action-wise Platt scaling, which learns separate recalibration maps per action dimension and then aggregates them. Empirically, this consistently outperforms global Platt scaling, temperature scaling, and histogram-based methods, and reveals that different degrees of freedom can be miscalibrated in different ways.
> >
> > More broadly, our work is in the same spirit as (possibly the most) influential deep learning UQ papers such as Guo et al. (2017) [1], Lakshminarayanan et al. (2017) [2], and Gal & Ghahramani (2016) [3]: the novelty lies not in inventing entirely new mathematical machinery and more in (i) identifying a previously unstudied but important setting (VLA calibration), (ii) adapting and extending simple tools in a way that respects its structure (instruction-level ensembles and action-wise recalibration), and (iii) uncovering new empirical phenomena (e.g., improved calibration mid-trajectory and heterogeneous calibration across action dimensions) that we hope will guide future, more sophisticated methods.
> >
> > - [1] On Calibration of Modern Neural Networks (https://arxiv.org/abs/1706.04599) - Introduces temperature scaling, a simple extension of Platt Scaling to a multi-class setting.
> > - [2] Simple and Scalable Predictive Uncertainty Estimation using Deep Ensembles (https://arxiv.org/abs/1612.01474) - show that a straightforward combination of standard tools (proper scoring rules, independent ensembles, and adversarial training) yields strong predictive uncertainty.
> > - [3] Dropout as a Bayesian Approximation: Representing Model Uncertainty in Deep Learning (https://arxiv.org/abs/1506.02142) - Show that dropout at test time yields well-behaved predictive uncertainties with minimal changes to training or architecture.
> >
> > **The claim of a systematic study is not well-supported by the experiments.**
> >
> > In response to this concern, we have:
> >  - Removed any potential overclaims (in the main paper, 1 occurrence of “systematic” and 1 occurrence of “comprehensive”).  Instead, we highlight that our study is the first-of-its-kind, as acknowledged by the reviewers.
> >  - Added experiments with 3 additional VLAs, and 1 additional (more difficult) task suite.  Please see comment above for more details on the new experiment additions and results.
> >
> > **All experiments are conducted on a single VLA, OpenVLA. This is not enough to claim a systematic study of calibration errors for VLAs. Diffusion-based VLAs might exhibit different calibration behavior.**
> >
> > We have substantially expanded the experiments in the revised draft to evaluate additional VLAs, strengthening the empirical scope of our study. In addition to OpenVLA, we now include three further VLAs with distinct architectures (MolmoAct, UniVLA, and NORA) and we add the LIBERO-10 suite for all four models. This yields 16 unique model/task-suite combinations (4 VLAs x 4 suites) plus 6 additional settings for 8-bit and 4-bit OpenVLA, for a total of 22 model/dataset combinations.
> >
> > We choose to focus on VLAs that produce discrete tokens with explicit logits, as this gives a natural notion of per-action probability and confidence. Models that predict continuous actions via flow matching (e.g., pi-0, pi0.5, FlowVLA, SmolVLA), diffusion (UnifiedVLA, GR00T N1), or L1 regression (VLA-Adapter, OpenVLA-OFT) would require additional methodological development to define and calibrate confidence, which we highlight as an interesting area for future work.

---

> ### Author Response · Authors · 2025-11-20
> **Rebuttal (ctd.)**
>
> **All experiments are conducted in the LIBERO simulation. For a study on this topic, the lack of any diverse simulation (or real-world) experiments is a significant weakness.**
>
> Studying calibration across a broader set of environments, including real robots, is an important direction. In this work we deliberately standardize on LIBERO because it is, to our knowledge, the dominant open-source benchmark for modern VLAs and the only one for which a wide range of fine-tuned models are publicly released. LIBERO is used to evaluate UnifiedVLA, FlowVLA, SmolVLA, OpenVLA, OpenVLA-OFT, VLA-Adapter, UniVLA, MolmoAct, Nora, pi-0, pi0-FAST, and pi-0.5, among others, whereas no other simulator appears in more than a small handful of these papers. Focusing on LIBERO lets us compare calibration behavior across multiple independently developed VLAs in a common, widely adopted setting.
>
> Regarding real-world experiments, our choice of simulation is driven by the sample complexity of calibration.  Our results feature measurements from more than 100,000 robot episodes.  On the other hand, prominent works from well-resourced research labs like Physical Intelligence (pi-0, pi-0.5) and Nvidia (GR00T N1) evaluate on the order of 100 real-world episodes.  This amount of data would not be enough to produce a single reliable ECE measurement.  We have added a further note about this to our limitations, pointing out that studying these techniques on real robots will ultimately be essential for understanding how various factors affect VLA calibration under realistic conditions.  Finally, we highlight that this choice of simulation-only evaluation is not uncommon in robotics research.
>
> **The paper's default choice of using the pre-action confidence (ci,1​) as the primary metric is not well-justified. OpenVLA is only trained on single-image observations and produces a single action. It has no explicit understanding of task progress. Thus, it is unclear if the confidence at the very first step is representative of the entire trajectory's confidence. Further ablations are required to validate this choice.**
>
> We agree that the choice of where to extract a scalar confidence from a VLA is important, and to our knowledge this is the first work that attempts to answer this question.  Our choice was made based on:
>  - Conventions from related domains like computer vision and LLMs (i.e., top-token probability).
>  - Domain-specific constraints; taking the estimate before first action is a universal choice available in any context, even the most safety-critical, whereas anything else is limiting in some contexts where even the first gradual action brings significant risk.
>
> It is true that there is a mismatch between what this score is explicitly measuring and what we want to estimate confidence in.  This is similar to the mismatch in LLMs between token probabilities and task success, as highlighted and exploited by methods such as Semantic Uncertainty [4], yet token probabilities are still accepted as a universal and reasonable baseline.  It is not clear what other choice would be as universally applicable and effective.
>
> In terms of validation of this confidence estimate as a reasonable choice, our experiment section contains many non-trivial calibration error measurements produced using this score.  These measurements would look very different if this score was not related to task success.  While we believe and hope that better confidence estimation methods may be developed in the future, we believe that we made a sensible choice, and that our experiments validate that this score is in fact related to ultimate task success.
>
> [4] Semantic Uncertainty: Linguistic Invariances for Uncertainty Estimation in Natural Language Generation (https://arxiv.org/abs/2302.09664)

---

> > ### Author Response · Authors · 2025-11-20
> > **Rebuttal (ctd.)**
> >
> > **Other aggregation methods are not evaluated or ablated.  Did you experiment with other aggregation metrics?**
> >
> > Yes, we do experiment with and compare multiple aggregation methods. Section 3.3 of the paper is devoted to this question of how different aggregation rules might affect calibration. In addition to the current step confidence, we evaluate a sliding window over recent estimates as well as an average of all estimates up to that timepoint. If the reviewer has any particular questions about these results or other potential approaches, we would be happy to discuss further.
> >
> > **What does the calibration look like in OOD scenarios? e.g., with new objects or unseen settings? What does the confidence look like without model fine-tuning? No experiments on zero-shot or out-of-domain environments, and the relation to model confidence.**
> >
> > We agree that these questions around zero-shot and OOD calibration are quite interesting.  We tried applying the OpenVLA model zero-shot, as well as the fine-tuned models OOD (e.g., evaluate Spatial model on Object, etc.).  We found 0% task performance across all zero-shot and OOD experiments, with trajectories showing catastrophic failures (the robot does not meaningfully attempt the task), which makes episode-level calibration statistics essentially ill-defined and uninformative. For this reason we focused our analysis on regimes with non-trivial success rates.  We have added a note to our Future Work section explicitly highlighting zero-shot/OOD calibration as a key avenue for future work.
> >
> > **Why do you compare against quantized versions of the same model?**
> >
> > We include 8-bit and 4-bit OpenVLA primarily as a robustness check: these quantized variants may be common in deployment, and we wanted to verify that our qualitative conclusions (e.g., the relationship between task error and calibration, and the gains from prompt ensembles and action-wise Platt scaling) persist under quantization. In the revised draft, we additionally add three other recent VLAs, so the quantized OpenVLA results now serve as supplemental evidence that our findings are stable.

---

> > > ### Author Response · Authors · 2025-11-26
> > > **rebuttal follow up**
> > >
> > > Hello, we just wanted to politely follow up here, and see if the reviewer has any remaining questions that we might be able to address.  We hope that our addition of results from 3 new VLAs (as well as 1 additional task suite) was able to address the reviewer’s main concerns.
> > >
> > > Thank you again for taking the time to review our submission.

---

### Meta-Review · Area_Chair_xEMn · 2026-01-06

**Summary:**

The reviewers have concerns around the evaluation (single VLA model and 3 LIBERO task suites) and how general the method is. During rebuttal the authors partially addresses the concerns (more VLA models, 1 more LIBERO task suites), but there are still remaining concerns not addressed. (no real world exp, no other arch such as pi0)

**Reviewer Concerns:**

The main concern is that the initial submission evaluated only a single VLA model (OpenVLA) on 3 LIBERO task suites. Te authors partially address this by adding 3 additional VLA model families (4 in total) and one additional task suite across all models. However, no real robot setup is studied.

There are also concerns regarding how general the method is. The authors did not provide real world exp results. The authors also did not provide study on different architectures such as pi0. There are several differences for modern VLAs such as 1. modern VLAs predict a trajectory instead of a single action. 2. modern VLAs have action tokenizers instead of predicting raw actions. It seems the study and proposed methods in this paper is not general enough.

**Reviewer Scores:**

Reviewer Cqir (2->2)
Does not fully address the concern on evaluation and generalization.

Reviewer EuHf (6->6)
no real robot set up.

Reviewer 5yfq (4->4)
Does not fully address the concern on evaluation and generalization.

Reviewer tjhn (8->8)
Minor issues are addressed

---

### Decision · Program_Chairs · 2026-01-26

Reject